# AUTOMATED PROOF GENERATION FOR RUST CODE VIA SELF-EVOLUTION

**Tianyu Chen** [1, *] **Shuai Lu** [2,†] **Shan Lu** [2] **Yeyun Gong** [2] **Chenyuan Yang** [3,*] **Xuheng Li** [4,*]
**Md Rakib Hossain Misu** [5,*] **Hao Yu** [1] **Nan Duan** [2] **Peng Cheng** [2] **Fan Yang** [2]
**Shuvendu K Lahiri** [2] **Tao Xie** [1] **Lidong Zhou** [2]
[1] Peking University, `{tychen811,yh0315,taoxie}pku.edu.cn`
[2] Microsoft Research, `{shuailu,shanlu,yegong,nanduan,pengc,fanyang,`
`shuvendu.lahiri,lidongz}@microsoft.com`
[3] University of Illinois at Urbana-Champaign, `cy54@illinois.edu,`
[4] Columbia University, `xuheng@cs.columbia.edu`
[5] University of California Irvine, `mdrh@uci.edu`

## ABSTRACT

Ensuring correctness is crucial for code generation. Formal verification offers a definitive assurance of correctness, but demands substantial human effort in proof construction and hence raises a pressing need for automation. The primary obstacle lies in the severe lack of data—there is much fewer proofs than code snippets for Large Language Models (LLMs) to train upon. In this paper, we introduce SAFE, a framework that overcomes the lack of human-written proofs to enable automated proof generation of Rust code. SAFE establishes a self-evolving cycle where data synthesis and fine-tuning collaborate to enhance the model capability, leveraging the definitive power of a symbolic verifier in telling correct proofs from incorrect ones. SAFE also re-purposes the large number of synthesized incorrect proofs to train the self-debugging capability of the fine-tuned models, empowering them to fix incorrect proofs based on the verifier's feedback. SAFE demonstrates superior efficiency and precision compared to GPT-4o. Through tens of thousands of synthesized proofs and the self-debugging mechanism, we improve the capability of open-source models, initially unacquainted with formal verification, to automatically write proofs for Rust code. This advancement leads to a significant improvement in performance, achieving a 52.52% accuracy rate in a benchmark crafted by human experts, a significant leap over GPT-4o's performance of 14.39%.

## 1 INTRODUCTION

Large Language Models (LLMs) have recently exhibited impressive capabilities in code generation (Roziere et al., 2023; Guo et al., 2024; Lozhkov et al., 2024; Google, 2024). However, the correctness of generated code cannot be guaranteed. To prove that a program satisfies the desired properties under *all* possible inputs, without running the program, we need *formal verification*. Unfortunately, formal verification is difficult to conduct, as one needs to first formally express the desired properties, often in a special proof-oriented language, and then craft formal proofs, taking substantial formal verification expertise (Zhang et al., 2024). Therefore, to advance trustable code generation, automated formal verification, particularly automated proof generation, is in a pressing need.

Two main approaches of proof automation have been explored. The first (Misu et al., 2024; Sun et al., 2024a; Pei et al., 2023; Liu et al., 2023; Chakraborty et al., 2023; Kamath et al., 2023; Chakraborty et al., 2024; Yao et al., 2023; Yang et al., 2024a) relies on well crafted prompt or in-context learning on LLMs. To teach LLMs about formal proofs, this approach often requires intensive prompt engineering that hardcodes proof-writing tips, and/or requires support from static program analysis, resulting in a limited generalization ability. The second approach (Polu et al., 2022; First et al., 2023; Azerbayev et al., 2023; Yang et al., 2024b) fine-tunes open-source LLMs.

---

* Work done during internship at Microsoft Research.
† corresponding author.

This approach has been shown to be effective for Lean (Avigad, 2017) (for mathematical proving) and F* (Swamy et al., 2016) (a special proof-oriented language), where tens of thousands of human-written Lean and F* proofs exist. However, how to apply this data-oriented approach to many other verification tools or proof languages that have much fewer exiting proofs remains as an open question.

In this paper, we tackle this open question for Verus (Lattuada et al., 2023), the state-of-the-art verification tool for code written in Rust. Verus is a perfect target for us, since (1) it is one of the rare verification tools that can directly prove the correctness of code written in a popular language (Rust), which is increasingly popular in production as a safer and better alternative to C/C++ (ONCD, 2024; Rivera et al., 2021); and (2) due to its short history, there are fewer than 500 Rust files currently verified by Verus on GitHub, too few for fine-tuning.

The challenge of lacking data for Verus proof generation shows up at three main levels. First, lack of suitable programs to quickly generate proofs for. Not all Rust features are supported by Verus yet. Furthermore, large Rust projects typically require code refactoring to get verified (Zhou et al., 2024; Sun et al., 2024b), and yet small Rust programs may not have meaningful properties to prove (e.g., a hello-world function). Second, lack of Verus specifications. Before proving any Rust function, Verus needs a formal specification that describes the expected behavior of the function, such as the pre-condition (Lines 7–9) and post-condition (Lines 10–13) shown in Listing 1. Not surprisingly, these specifications do not exist except for in those existing verified Rust files on GitHub. Third, lack of Verus proofs. It took many months to write a couple of thousand lines of Verus proofs even for experts (Zhou et al., 2024; Sun et al., 2024b). It took more than ten years' effort to accumulate those fine-tuning training data for Lean (Yang et al., 2024b) and F-Star (Chakraborty et al., 2024).

In this paper, we propose the **S**elf-evolving **A**utomated proo**F** g**E**neration (**SAFE**) framework that overcomes the challenge of multi-level data scarcity. First, to quickly obtain a large number of proof-friendly Rust programs, we leverage GPT-4o to "translate" tens of thousands of small Python and Rust programs in popular code-synthesis datasets to programs written in Verus-supported Rust syntax. Any program that cannot be compiled by the Verus compiler is dropped.

Second, to obtain formal specifications for these tens of thousands of Rust programs, SAFE uses a few rounds of self-evolving specification generation. The bootstrapping round prompts GPT-4o to generate Verus-style specifications based on each Rust function and its associated natural language doc-string. In every following round, specifications generated in previous iterations are used to train a new fine-tuned open-source LLM that produces more specifications in generally higher quality for the next round of finetuning. Keys to the success of this step include (1) using a quantitative metric (Lahiri, 2024) to differentiate high-quality specifications from low-quality ones, and use the former for only finetuning; and (2) the observation that we need only *reasonably well*, instead of *perfect*, specifications to enable proof generation in the next step.

Next, to obtain tens of thousands of verified Rust programs, SAFE similarly uses a self-evolving procedure. Code proofs are very difficult to synthesize: even with careful prompt engineering, GPT-4o managed to synthesize correct proofs for only $< 20\%$ of programs in the the preceding dataset with synthesized specifications, after a month of non-stop invocations. Fortunately, this is sufficient to bootstrap SAFE. Through rounds of self-evolving, the quantity and quality of synthesized proofs keep increasing, while fine-tuned open-source models' capability of proof synthesis keeps getting augmented. These models also generate proofs much faster than GPT-4o. The keys to the success here include (1) the ability of Verus in authoritatively and quickly telling correct proofs from incorrect ones, allowing SAFE to sift through the huge amount of low-quality data in early rounds *without* any human labeling, and (2) the reasonable quality of SAFE-synthesized specifications that allow SAFE to largely avoid trivial proofs, which have little usage in fine-tuning.

Finally, SAFE re-purposes the huge number of *incorrect* proofs generated in the previous step to add the self-debugging capability into its model. In each round, when a correct proof $P_\checkmark$ is synthesized after several attempts, the triplet of an earlier incorrect proof $P_X$, the verification error reported by Verus on $P_X$, and $P_\checkmark$, becomes a training data point, which fine-tunes the model's capability in debugging and repairing proof—a great side effect of having incapable models early on.

In our experiments, SAFE leverages DeepSeekCoder (Guo et al., 2024) as the generator, successfully synthesizing 19,017 formal specifications and 9,706 verified Rust functions, from a dataset comprising 45,395 Rust functions sourced from the MBPP (Austin et al., 2021) training split and

Listing 1: An Example Verus Program (Binary Search).

```
1   verus!{
2   // Performs a binary search on a sorted vector of 64-bit unsigned integers (u64) to find the
    index of a given target value.
3   fn binary_search(v: &Vec<u64>, k: u64) -> (r: usize)
4       requires //pre-conditions of this program
5           forall|i:int, j:int| 0 <= i <= j < v.len() ==> v[i] <= v[j],
6           exists|i:int| 0 <= i < v.len() && k == v[i],
7       ensures //post-conditions of this program
8           0 <= r,
9           r < v.len(),
10          k == v[r as int],
11  {
12      let mut i1: usize = 0;
13      let mut i2: usize = v.len() - 1;
14      while i1 != i2
15          invariant //loop invariants (used for proof)
16              i2 < v.len(),
17              exists|i:  int| i1 <= i <= i2 && k == v[i],
18              forall|i:  int, j:  int| 0 <= i <= j < v.len() ==> v[i] <= v[j],
19      {
20          let ix = i1 + (i2 - i1) / 2;
21          if v[ix] < k {
22              i1 = ix + 1;
23          } else {    i2 = ix;    }
24      }
25      i1
26  }}
```

This Rust program's pre-conditions are highlighted in orange background; its post-conditions are highlighted in green; and its Verus proof annotations (loop invariants) are highlighted in gray.

the CodeNet (Puri et al., 2021) training dataset. Note that the initial Rust dataset contains **zero** lines of formal specification or proof. Our evaluation on a human-curated Verus benchmark with human-written specifications, **VerusBench**, and a synthetic benchmark, **CodeNet-Test**, demonstrates that SAFE empowers DeepSeekCoder, which is initially unacquainted with Verus, to achieve 43.17% and 43.83% accuracy on the two benchmarks by direct generation, far surpassing GPT-4o's performance of 11.51% and 0.28%, respectively. Furthermore, the model's accuracy reaches 79.14% in VerusBench and 48.43% in CodeNet-Test once its self-debugging feature is used.

## 2   BACKGROUND AND RELATED WORK

### 2.1   VERUS VERIFICATION TOOL

Verus (Lattuada et al., 2023) statically analyzes every proof target (i.e., a Rust function and its specification) and any given proof annotations, and forms queries for the underlying SMT solver (e.g., Z3 (De Moura & Bjørner, 2008)) to solve. By leveraging the type system in Rust and allowing both specification and proof annotations to be written in Rust syntax, Verus has become the state-of-the-art verification tool for Rust, one of the most popular programming languages (Perkel, 2020; Fulton et al., 2021), and has been used to successfully verify large systems (Sun et al., 2024b; Zhou et al., 2024). Unfortunately, since Verus has been developed for only about three years, there are fewer than 500 Verus verified Rust files on GitHub.

Listing 1 shows a Rust function that implements binary search. Its specification includes a pre-condition enclosed in a `requires` block and a post-condition enclosed in an `ensures` block (Lines 4-10). The pre-condition states that the input array is ordered in an ascending way and the input value `k` to be searched exists in the array (`forall` and `exists` are Verus quantifiers); the post-condition requires that the return value $r$ should be a valid index pointing to the input value `k`.

For simple programs/specifications, Verus can accomplish the formal verification without any extra annotations (we refer to these cases as *trivial proofs*). Unfortunately, for functions and specifications that involve loops, collections, and quantifiers, Verus often needs users to provide *proof annotations*. The loop invariants specified on Lines 16–18 are one type of proof annotations. They specify what properties are true right before and right after every loop iteration. Verus will prove the correctness of each loop invariant and then use proved loop invariants to help prove the function specification. If

any loop invariant is incorrect or if needed invariants are missing, the proof will fail. For more complicated tasks, other types of proof annotations like `assert` and lemma functions may be needed.

## 2.2 Prior work in proof-benchmark building and self-evolving framework

There has been much effort recently in building datasets of existing human written proofs (Loughridge et al., 2024; Zhang et al., 2024; Chakraborty et al., 2024). Unfortunately, since writing proofs takes expertise beyond normal coding, this dataset-building approach is difficult to scale. Recent work has explored augmenting the Dafny-proof dataset using proofs synthesized by GPT-4 through few-shot learning and human-written proof examples (Misu et al., 2024; Sun et al., 2024a). However, due to the limited proof-generation power of GPT-4, the number of proofs in these datasets are fewer than 200, with many being trivial proofs.

Self-evolving style of learning has been explored in other contexts: A reinforced self-training framework is proposed in the machine translation task (Gulcehre et al., 2023), and the expert iteration strategy is leveraged for math proving with Lean (Polu et al., 2022). Our work applies self-evolving and expert iteration to synthesize proofs for Rust code. This new task raises different challenges from prior tasks like writing math proofs (Polu et al., 2022) and hence requires different designs.

One obvious challenge is the issue of data scarcity discussed in Section 1: SAFE does not have access to billions of tokens of manually-written examples (e.g., math proofs) and hence cannot rely on fine-tuned models for bootstrapping as in prior work; in fact, SAFE does not even have access to a large quantity of proof problems and hence has to synthesize problems by itself (i.e., Verus-compatible Rust functions and the associated specifications).

Another major challenge is related to the underlying verification engine. Supported by an interactive theorem prover, Lean (Avigad, 2017), math-proof synthesis is naturally decomposed to many small steps or tactics, with clear judgment about what are useful intermediate proofs. In contrast, like many code-verification tools (Leino, 2010; Swamy et al., 2016), Verus leverages a SMT solver to prove the correctness of a function *as a whole*, while proof annotations are used as hints to the SMT solver. It is difficult to decompose Verus-proof synthesis into small steps, and it is difficult to judge whether incorrect proof annotations are useful. Consequently, step-wise search strategies used in prior work do not apply here. Instead, whole-proof debugging is crucial for SAFE.

We believe that SAFE presents a new application of the self-evolving and expert iteration philosophy to an important and challenging task—code-proof automation. Through its automated measuring and filtering at every step of data synthesis—Verus compiler for program transpilation, the quantitative specification-quality metric for specification synthesis, and Verus verification for proof-synthesis/debugging, SAFE overcomes those unique challenges and allows high-quality training data to gradually accumulate without manual effort.

## 3 Approach

As illustrated in Figure 1, SAFE involves two self-evolving (Tao et al., 2024; Gulcehre et al., 2023) procedures. The first procedure synthesizes specifications that are used as inputs to the second procedure, while the second procedure produces the end-result of SAFE—a fine-tuned LLM that can automatically synthesize proofs for Rust code. As listed in Algorithm 1, both procedures use GPT-4o to generate the round-0 data $spec/proof\text{-}data_0$. In each round $r$, we use data collected and filtered from earlier rounds, $0...r-1$, to fine-tune $model_r$ based on its preceding $model_{r-1}$. At the end of each round, $model_r$ is used to generate the round-$r$ data. High-quality specifications are preserved for the next round. Low-quality specifications are discarded; incorrect proofs are re-purposed to become self-debugging training data for the next round.

### 3.1 Step 1: Generating Verus-compatible Code

This step ensures the compatibility of our input Rust programs with Verus, which does not support all Rust features. For instance, some Rust expressions and standard library functionalities like `for`, `Iterators`, `HashMap`, and others are not supported or only partially supported by Verus. Thus, it is necessary to adapt normal Rust code to Verus-compatible one before adding specifications.

To address this compatibility issue, we employ GPT-4o as a code translator, effectively substituting Verus-incompatible Rust code snippets with Verus-compatible alternatives, such as converting

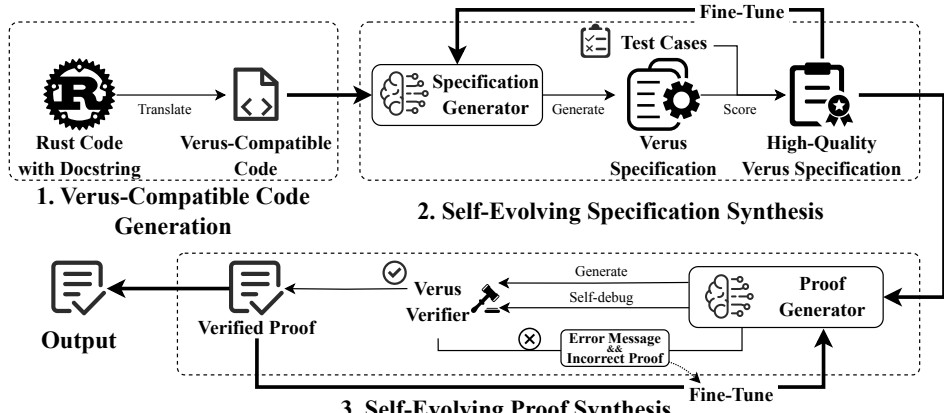

Figure 1: The SAFE Framework

---

**Algorithm 1:** Self-Evolving

**Input** : $model_0$, an open-source LLM; $rust\_programs$, a set of Rust programs
**Output:** $model_r$, the self-evolved LLM

1  $programs \leftarrow$ GPT-4o.$translate(rust\_programs).compilable$(Verus Compiler);
2  **Specification Generation:**
3     $spec\text{-}data_0 \leftarrow$ GPT-4o.$generate(programs).filter(Tests)$;
4     **for** $r$ $in$ $[1, \ldots R_1]$ **do**
5        $model_r \leftarrow model_{r-1}.fine\text{-}tune(\bigcup_{i=1}^{r-1} spec\text{-}data_i)$;
6        $spec\text{-}data_r \leftarrow model_r.generate(programs).filter(Tests)$;
7     **return** $model_r$;

8  **Auto-Proving:**
9     $proof\text{-}data_0, debug\text{-}data_0 \leftarrow$ GPT-4o.$generate(programs).filter$(Verus Verifier);
10    **for** $r$ $in$ $[1, \ldots R_2]$ **do**
11       $model_r \leftarrow model_{r-1}.fine\text{-}tune(\bigcup_{i=1}^{r-1} (proof\text{-}data_i + debug\text{-}data_i))$;
12       $proof\text{-}data_r \leftarrow model_r.generate(programs).filter$(Verus Verifier);
13       $debug\text{-}data_r \leftarrow model_r.debug(programs).filter$(Verus Verifier);
14    **return** $model_r$;

---

an iterator-based implementation into a while-loop based implementation. Specifically, we prompt GPT-4o with all the unsupported features sourced from Verus official documentation. All the converted code that can pass the grammar check of Verus compiler is collected for next steps. This step is performed only once in our self-evolving framework.

### 3.2 STEP 2: SELF-EVOLVING SPECIFICATION SYNTHESIS

In this step, LLMs are tasked with generating preconditions and postconditions for a Rust function based on the function's implementation and docstring. Different from prior work in specification generation (Flanagan & Leino, 2001; Ma et al., 2024), SAFE needs a synthesized specification as input for its self-evolving proof-synthesis framework, which raises unique requirements on its specification evaluation criteria (i.e., which specification to keep or discard) and the mechanism to evaluate the criteria.

In terms of criteria, a perfect specification $S$ should be *correct* (i.e., any correct implementation should be accepted by $S$) and *complete* (i.e., any incorrect code implementation should be rejected by $S$). In SAFE, getting perfect specification is **not** the goal. SAFE should discard incorrect specifications, which can never be proved and hence will waste training cycles in the next step. However, SAFE is fine with incomplete specifications, as the usage of specifications in SAFE is to stimulate

proof synthesis not to judge code correctness. In fact, SAFE needs incomplete specifications, as complete specifications are likely too challenging to prove at the early stage of the proof-synthesis model's evolution. Of course, specifications that are too incomplete should be discarded, because trivial specifications can be proved without any proof annotations and hence offer no usage for proof-synthesis training (e.g., `0 <= r` for the binary search function).

In terms of the mechanism to evaluate the preceding criteria, previous work relies on running many test cases (Endres et al., 2024), formally verifying the consistency between specifications and code (Ma et al., 2024), or user inspection (Lahiri et al., 2022). None of these mechanisms suit SAFE: user inspection or running many test cases would severely slow down the self-evolving framework; formal verification also does not work when we do not have a capable model to automatically synthesize proofs.

With all these constraints, SAFE chooses to leverage a recently proposed technique (Lahiri, 2024) to measure the *Correctness* and *Completeness* of specifications. We use Verus to symbolically evaluate what percentage of test cases $\mathbf{T}$ provided by the original dataset can pass a given specification $S$ (i.e., the *Correctness* score), and what percentage of test cases $\mathbf{T}'$ mutated from $\mathbf{T}$ can be rejected by $S$ (i.e., the *Completeness* score). For a given method $m(x) : y$, with $x$ being the input parameter and $y$ being the output, we denote the input/output-value pair of any test case in a test suite $\mathbf{T}$ as $(i, o)$ and generate a corresponding mutated test case by changing $o$ into a randomly different value $o'$. The two metrics for a specification $S$ are formally defined as following, with $S(i, o)$ being a boolean value representing whether the input-output value pair $(i, o)$ satisfies the specification $S$ or not:

$$\begin{aligned}
\mathbf{Correctness}_S: &\ \frac{|\ \{\ (i, o)\ |\ S(i, o)\ \ and\ \ (i, o) \in \mathbf{T}\}\ |}{|\ \mathbf{T}\ |} \\
\mathbf{Completeness}_S: &\ \frac{|\ \{\ (i, o')\ |\ not\ S(i, o')\ \ and\ \ (i, o') \in \mathbf{T}'\}\ |}{|\ \mathbf{T}'\ |}
\end{aligned} \tag{1}$$

We can now set a threshold that fits the need of SAFE: we keep all the specifications with $\geq 80\%$ *Correctness* score and $\geq 60\%$ of *Completeness* score. This way, SAFE can obtain many imperfect but useful specifications for its proof synthesis. We do not set the Correctness threshold to be 100%, because we use GPT to generate test cases when few test cases are provided by the dataset, and we cannot guarantee perfect correctness of these test cases. Since Verus can symbolically evaluate $S(i, o)$ in the preceding formula without checking or running the Rust function, the scoring is fast.

For each function, we keep up to three synthesized specifications, which not only enrich the specification training dataset but also increase the chance of synthesizing valid proofs later. Our self-evolving process for specification stops when there is no significant increase in the total number of accepted function–specification pairs, at which time all these pairs move on to become the input for the next step.

### 3.3 STEP 3: SELF-EVOLVING PROOF SYNTHESIS

Now that we have tens of thousands of function-specification pairs, we can start the self-evolving procedure of proof synthesis. Compared with specification synthesis, deciding whether to accept or reject a synthesized item is much more straightforward—only proof that Verus can use to verify the Rust function satisfies its specification is accepted; if any verification error is raised, the proof is discarded[1]. However, since proof synthesis is much more difficult than specification synthesis, we need to pay special attention to bootstrap the self-evolving procedure and to keep it moving forward.

For bootstrapping, a simple prompt to GPT-4o would not work, as it is very difficult for GPT-4o to synthesize all the needed loop invariants without any incorrect ones in between, not to mention more complicated proof annotations like proof blocks and lemma functions. Therefore, we write a detailed prompt for GPT-4o, explaining the principles and tricks of writing Verus proofs. By doing so, we finally manage to make GPT-4o synthesize proof annotations that allow Verus to prove more than one thousand Rust functions at the cost of one whole month of non-stop GPT-4o invocations. Fortunately, doing so is sufficient for bootstrap. For the remainder of the self-evolving procedure, GPT-4o is not used any more. Instead, we use an open-source LLM much smaller than GPT-4o to efficiently synthesize data and improve its capability in a self-evolving manner as in Algorithm 1. Specifically, we fine-tune our open-source LLM on two tasks, proof generation and self-debugging.

---

[1] If a Rust function is in fact buggy or a synthesized specification is incorrect, no proof will be accepted.

**Task #1: Proof generation.** The synthesized proofs and specifications are directly used for training this task. Taking a code snippet with specifications as input $S$, LLM $M$ is tasked with generating a proof $Y$ (consisting of tokens $y_0, \ldots, y_N$) that can make the code proved by Verus. We train this task with a sequence-to-sequence objective where

$$\mathcal{L}_{Gen}(\theta) = -\sum_{i=0}^{N} \log P_\theta(y_i \mid S, y_{t<i}) \tag{2}$$

**Task #2: Self-debugging.** The self-debugging task is trained on the data pair of an incorrect proof and its revision. "Thanks to" the incapability of LLMs in proof generation, a huge number of incorrect proofs are generated during data synthesis. For every Verus program (together with its specification) for which a perfect proof $Y_\checkmark = (y_0, ..., y_N)$ is eventually synthesized in a round, we denote those incorrect proofs generated before $Y_\checkmark$ as $[Y_1, \ldots, Y_m]$. For every incorrect proof $Y_X \in [Y_1, \ldots, Y_m]$, a triplet $\{Y_X, \texttt{Error}_{Y_X}, Y_\checkmark\}$ is added to our self-debugging training dataset, where $\texttt{Error}_{Y_X}$ represents the verification errors reported by Verus about $Y_X$. The objective of self-debugging task is

$$\mathcal{L}_{Debug}(\theta) = -\sum_{i=0}^{N} \log P_\theta(y_i \mid Y_X, \texttt{Error}_{Y_X}, y_{t<i}) \tag{3}$$

It is important to train the model to "debug" an imperfect proof. In many cases, the proof annotations generated by the model contain only small errors or miss one line of annotation. Without the capability of self-debugging, the model would start from scratch and fail to synthesize a perfect proof even after many attempts. In SAFE, we conduct joint training for the proof generation task and the self-debugging task, empowering the LLM to both generate a proof from scratch and repair an existing proof based on verification error messages for a Rust function and its specification.

## 4 EXPERIMENTS

### 4.1 DATASET

As an early exploration of automated formal verification, in this paper, we focus solely on the Rust code of algorithm types at the function level, as our source of data. We employ the MBPP dataset (Austin et al., 2021) (only training split for data synthesis) and the CodeNet dataset (Puri et al., 2021) as our data sources. These two datasets contain small programs written in different programming languages, each of which is an intended solution to a coding problem described in natural language and is associated with three test cases on average. We translate the Python programs in MBPP and extract the Rust programs in CodeNet. In total, we collect 45,395 Rust single-function programs. Of these programs, 21,398 have been successfully transformed into Rust code that is compatible with Verus by GPT-4o. We conduct the specification synthesis first, and after two rounds of self-evolution, we obtain 19,017 high-quality specifications. Then we run three rounds of self-evolving proof synthesis, ending with 9,706 verified programs, and 10,486 self-debugging data pairs. To the best of our knowledge, this dataset represents the most extensive synthetic dataset created for Verus code to date.

### 4.2 BENCHMARK AND METRICS

We assess a model's proficiency in generating proofs in two benchmarks, the human-written benchmark **VerusBench** and the synthetic benchmark **CodeNet-Test**. **VerusBench** is a human-written benchmark dataset with human-written Rust code and corresponding Verus specifications. It contains 139 code files in total. 23 of them are algorithmic programs from Verus tutorials. 38 of them come from the dataset of SV-COMP-2021 (Beyer, 2021) (in short, SV), which is a contest focused on program verification for C and Java languages; we utilize 38 Verus-translated tasks provided by Yao et al. (2023). The remaining 78 tasks come from MBPP-DFY-153 (Misu et al., 2024), a Dafny version of the MBPP test set. We have translated 78 of them that are compatible with Verus. **CodeNet-Test** is an expansive benchmark, 10x larger than **VerusBench**, crafted by LLM and encompassing 1,435 diverse tasks. Given the substantial human effort required to write Verus code and specifications, it's impractical to create a large test suite by human. To comprehensively measure the model capability of generating proofs, we split a subset of CodeNet for testing, ensuring that it is not utilized in our data synthesis process. We leverage our specification generator to craft specifications for this subset and apply the same scoring mechanism to preserve *reasonably well* specifications.

Table 1: Accuracy of SAFE and baselines. Prompt is GPT-4o with a long prompt used to bootstrap SAFE; column SAFE is LLaMa3.1 or DeepSeekCoder with three rounds of finetuning and a simple prompt. For SAFE+, Accuracy@2 means getting one initial proof sample and one debugging sample; Accuracy@100 means getting 10 initial proofs and then $10 \times 10$ debugging samples.

| Benchmark | Accuracy | GPT-4o | | LLaMa3.1 | | | DeepSeekCoder | | |
|---|---|---|---|---|---|---|---|---|---|
| | | Raw | Prompt | Raw | SAFE | SAFE+ | Raw | SAFE | SAFE+ |
| VerusBench | @1 | 11.51 | 25.90 | 3.60 | **46.04** | - | 9.35 | 43.17 | - |
| | @2 | 14.39 | 30.93 | 5.76 | 48.20 | **52.52** | 10.79 | 46.76 | 49.64 |
| | @10 | 24.46 | 41.01 | 11.51 | **53.96** | - | 17.27 | **53.96** | - |
| | @100 | 43.88 | 46.76 | 28.78 | 55.40 | 64.03 | 32.37 | 59.71 | **70.50** |
| CodeNet-Test | @1 | 0.28 | 2.86 | 0.00 | **44.32** | - | 0.21 | 43.83 | - |
| | @2 | 0.70 | 3.41 | 0.03 | 45.34 | **48.50** | 0.55 | 44.74 | 48.43 |

We measure the capability of proof generation by Accuracy@K. Specifically, we use any model under evaluation to sample K proofs during proof generation. For Accuracy@1, we use greedy decoding, which generates only 1 output, and for Accuracy@10, we sample 10 outputs with a temperature of 0.7. Then, for each task in our benchmark set, Accuracy@K equals one if at least one proof/debugged proof is verified by Verus. Note that, when the self-debugging feature of SAFE is used (i.e., SAFE+), the proof generation includes two rounds: at the first round, $K$ initial proofs are sampled; at the second round, if none of the initial proofs are correct, $K$ proofs are generated by self-debugging *each* initial proof, resulting in K * K debugged proofs.

## 4.3 EVALUATION RESULTS

In Table 1, we compare the accuracy of SAFE with our baseline approaches. Due to the inherent complexity of this task, there is no pre-existing fine-tuned model to serve as a baseline. We employ GPT-4o, DeepSeekCoder-33B-Instruct model (Guo et al., 2024), and Llama3.1-8B-Instruct model (Dubey et al., 2024) with basic prompts **the same** as what we use for the fine-tuned models as our baselines. To offer some advantages to the baselines, we also feed four examples for in-context learning in these baselines, which we do **not** use for SAFE fine-tuned models.

We can see that SAFE achieves substantially higher accuracy compared to baseline approaches in both datasets and metrics. In the VerusBench benchmark, even without self-debugging, SAFE achieves an Accuracy@1 of 46.04% and an Accuracy@10 of 53.96% while the best of 'Raw' results come from GPT-4o with 11.51% in Accuracy@1 and 24.46% in Accuracy@10. With the long and carefully designed prompt, which was used to bootstrap SAFE proof synthesis, GPT-4o is reasonably effective for VerusBench. For the CodeNet-Test benchmark, we find the performance of baselines and prompt-based GPT-4o, denoted as Prompt, drops significantly—by more than $10X$ compared with VerusBench, while SAFE does not suffer as much. It is important to note that CodeNet-Test is significantly larger and exhibits a distinct data distribution compared to VerusBench, and the former is derived from competitive programming, whereas the latter encompasses common algorithms. The results indicate that the prompt-based approach has limited generalizability in automated proof generation, while SAFE no longer needs complicated prompts during inference. We further report paired t-test (Hsu & Lachenbruch, 2014) results for Table 1 in Table 5 in Appendix.

Table 1 also shows the efficacy of self-debugging (i.e., SAFE+). For example, after one initial proof is sampled, applying self-debugging on this proof to produce the second sample (Accuracy@2 for SAFE+) consistently outperforms simply producing a second sample of proof without debugging (Accuracy@2 for SAFE) for both LLaMa and DeepSeekCoder on both VerusBench and CodeNet-Test, as shown in Table 1. SAFE+ with DeepSeekCoder achieves the best performance of 70.50% in Table 1 when $10\times10$ debugged proofs are produced for 10 initial proofs for each proof task.

Overall, by leveraging our self-evolving framework, SAFE can substantially improve the capability of open-source model and effectively generate proofs for Verus programs.

### 4.3.1 BENEFITS FROM SELF-EVOLVING

In Table 2, we use the performance on VerusBench to show how our self-evolving framework improves the capability of models. In this set of experiments, we use DeepSeekCoder-33B-

Table 2: Accuracy of SAFE models produced by each round (measured on VerusBench)

| Self-Evolving | Metric | Proof Generation | | | | Self-Debugging ($K + K * K$) | | | |
|---|---|---|---|---|---|---|---|---|---|
| | | SV | MBPP | Tutorial | Total | SV | MBPP | Tutorial | Total |
| GPT-4o | Accuracy@1 | 39.47 | 19.23 | 26.09 | 25.90 | - | - | - | - |
| | Accuracy@10 | 60.53 | 33.33 | 34.78 | 41.01 | - | - | - | - |
| Round 1 | Accuracy@1 | 55.26 | 29.49 | 4.35 | 32.37 | 57.89 | 30.77 | 4.35 | 33.81 |
| | Accuracy@10 | 81.58 | 30.77 | 13.04 | 41.73 | 81.58 | 30.77 | 13.04 | 41.73 |
| Round 2 | Accuracy@1 | 73.68 | 29.49 | **34.78** | 42.45 | **84.21** | 38.46 | **39.13** | **51.08** |
| | Accuracy@10 | 89.47 | 30.77 | 47.83 | 49.64 | 92.10 | 56.41 | 52.17 | 65.47 |
| Round 3 | Accuracy@1 | **78.95** | **30.77** | 26.09 | **43.17** | 81.58 | **41.03** | 26.09 | 49.64 |
| | Accuracy@10 | **92.11** | **35.90** | **52.17** | **53.96** | **97.37** | **58.97** | **65.22** | **70.50** |

Instruct for data synthesis and fine-tuning; Rounds 1, 2, 3 represent three models fine tuned from DeepSeekCoder-33B-Instruct. Table 2 illustrates that for every subset of VerusBench, the best Accuracy@1 and Accuracy@10 scores are achieved at the last two rounds of model evolution, indicating the effectiveness of self-evolution. Besides, Round 1 model with a simple prompt already outperforms GPT-4o with a much more sophisticated prompt for all but the "Tutorial" subset. Round 3 model outperforms Round 2 model in terms of Accuracy@10 for all subsets of tasks, with or without self-debugging. Meanwhile, the improvement from Round 2 to Round 3 is much smaller than the improvement from GPT-4o to Round 1, and from Round 1 to Round 2, justifying our decision of stopping the self-evolution after three rounds. We further report paired t-test results for Table 2 in Table 6 in Appendix.

Finally, Round 3 model greatly outperforms GPT-4o for all three sub-sets of VerusBench, even though GPT-4o uses a much more sophisticated prompt: GPT-4o is most effective for the SV subset with a 39.47% Accuracy@1, while Round 3 model improves that metric to 78.95% (81.58% with self-debugging); GPT-4o is the least effective for the MBPP subset with 19.23% Accuracy@1, which is improved by Round 3 model to 30.77% (41.03% with self-debugging). This trend bolds well for SAFE to work with different datasets and different bootstrapping approaches in the future.

### 4.3.2 Improvement of Self-debugging

Table 1 demonstrates that SAFE+, which allows LLMs to do self-debugging, substantially improves the accuracy compared to direct generation, indicating the effectiveness of the self-debugging mechanism. As we show in Listing 2 of the Appendix, Verus provides error messages for any incorrect proof about which proof annotation, which part of the specification, or which implicit proof target (e.g., no overflow for every arithmetic expression) cannot be verified. These error messages can help repair the proof for experienced human users and well trained models.

**Decoding Strategies for Self-Debugging.** From Table 1, we can see that the enhancements of self-debugging become more significant with the increase in the number of sampled outputs. To further evaluate how decoding strategies affect the performance of self-debugging in VerusBench, we conduct four settings by combining greedy and sampling decoding during the generation and debugging phases, and run two more self-debugging rounds with greedy decoding. Figure 2 shows that more rounds of self-debugging hardly improve the accuracy, with the improvement less than 1%. Besides, the sampling decoding strategy is more useful for the generation phase rather than for the self-debugging phase. Notably, the "sampling+greedy" (generating multiple proofs and for each incorrect proof generate one debugged proof) strategy outperforms the "greedy+sampling" (generating one proof and generate multiple debugged proofs) setting. Existing work (Chen et al., 2023; Olausson et al., 2023) has shown similar results.

### 4.3.3 Impact of Specification Quality

As mentioned in Section 3.2, we keep only specifications with high Correctness scores and reasonably high Completeness scores. To investigate the impact of this design decision, we zoom into the last round of proof-synthesis model fine-tuning to see how the quality of a proof-synthesis model can be affected by the quality of the used specification dataset. Specifically, the current Round-3 SAFE model is obtained by fine-tuning the Round-2 SAFE model using correct proofs synthesized

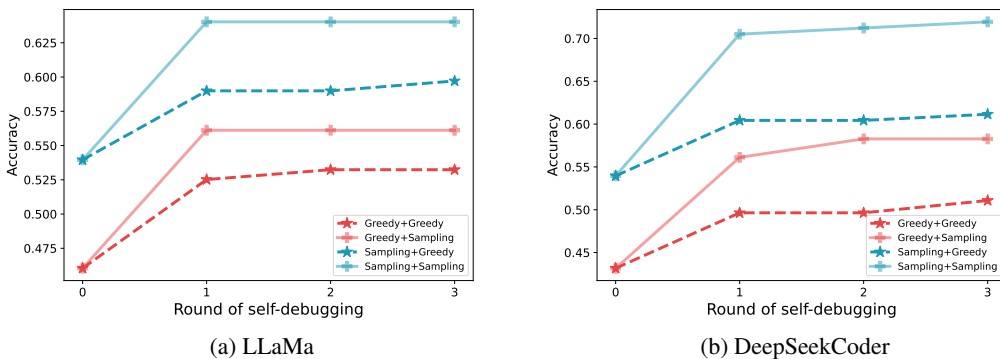

| (a) LLaMa | (b) DeepSeekCoder |

Figure 2: SAFE's accuracy with different self-debugging, sampling decoding strategies

Table 3: Accuracy drop of proof synthesis w/ training-specification quality drop (on VerusBench)

| Model | Metric | Proof Generation | | | | Self-Debugging ($K + K * K$) | | | |
|---|---|---|---|---|---|---|---|---|---|
| (Spec-Quality) | | SV | MBPP | Tutorial | Total | SV | MBPP | Tutorial | Total |
| SAFE | Accuracy@1 | 78.95 | 30.77 | 26.09 | 43.17 | 81.58 | 41.03 | 26.09 | 49.64 |
| | Accuracy@10 | 92.11 | 35.90 | 52.17 | 53.96 | 97.37 | 58.97 | 65.22 | 70.50 |
| Low-Quality | Accuracy@1 | 0.00 | 29.49 | 4.35 | 17.27 | 0.00 | 43.59 | 21.74 | 28.06 |
| | Accuracy@10 | 5.26 | 32.05 | 21.74 | 23.02 | 15.79 | 50.00 | 47.83 | 40.29 |
| Mix-Quality | Accuracy@1 | 36.84 | 30.77 | 17.39 | 30.22 | 42.11 | 43.59 | 21.74 | 39.57 |
| | Accuracy@10 | 65.79 | 32.05 | 30.43 | 41.01 | 71.05 | 51.28 | 52.17 | 56.83 |

by the bootstrapping model, the Round-1 model, and the Round-2 model based on the specifications $S$ selected by SAFE, denoted as $\mathbf{P_0}[\mathbf{S}] \cup \mathbf{P_1}[\mathbf{S}] \cup \mathbf{P_2}[\mathbf{S}]$ ($\mathbf{P}_i[\mathbf{S}]$ denotes all the correct proofs synthesized in round $i$ based on specification set $S$). In our two alternative settings, we go back to the set of all specifications synthesized during the specification synthesis self-evolving procedure and *randomly* sample a set of specifications $S*$ that have the same size as $S$; naturally $S*$ contains many low-quality specifications that do not pass our Correctness or Completeness threshold. In our alternative setting of 'Mix-Quality', we obtain a model by fine-tuning the Round-2 SAFE model using $\mathbf{P_0}[\mathbf{S}] \cup \mathbf{P_1}[\mathbf{S}] \cup \mathbf{P_2}[\mathbf{S}*]$; in our alternative setting of 'Low-Quality', we obtain a model by fine-tuning the original DeepSeekCoder using $\mathbf{P_2}[\mathbf{S}*]$ alone.

From the results in Table 3, we can see that high-quality specifications contribute substantially to the end-to-end effectiveness of SAFE. If a model is trained without any high quality data ('Low-Quality'), the accuracy drops for all three subsets of VerusBench and even drops to 0% Accuracy@1 and 5.26% Accuracy@10 for SV. Probably because more debugging data is created in this setting, the self-debugging feature helps more for this setting, but the overall accuracy still lags way behind the default SAFE Round-3 model. When low quality data is mixed with high quality one, the performance still drops as shown in the 'Mixed-Quality' row in the table. In Appendix, we further report paired t-test results for Table 3 in Table 7, and provide an example Rust program (Listing 14) whose low quality specification leads to an extremely simple proof—this simple proof probably offers no benefit for proof-synthesis fine-tuning. In conclusion, specification selection matters to the fine-tuning and the evolution of proof-synthesis models.

## 5 CONCLUSION

In this paper, we have proposed SAFE, a novel self-evolving framework to advance automated proof generation for Rust. SAFE alleviates the severe data scarcity challenge by coupling data synthesis and model fine-tuning in a self-evolving manner, demonstrating superior efficiency and precision compared to relying solely on GPT-4o. Through ten of thousands of synthesized proofs and the self-debugging mechanism, we improve the capability of open-source models to automatically write proofs for Rust code. Our evaluation shows that SAFE has a significant improvement over GPT-4o.

## ACKNOWLEDGMENTS

This work was partially supported by National Natural Science Foundation of China under Grant No. 92464301. Tao Xie is also affiliated with the School of Computer Science, Peking University; Key Laboratory of High Confidence Software Technologies (Peking University), Ministry of Education, Beijing, China

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

APPENDIX

## A  VERUS EXAMPLES

Listing 2: An imperfect proof of binary search with its error message recieved from Verus.

```
1   use vstd::prelude::*;
2   fn main() {}
3
4   verus!{
5   // Performs a binary search on a sorted vector of 64-bit unsigned integers (u64) to find the
    index of a given target value.
6   fn binary_search(v: &Vec<u64>, k: u64) -> (r: usize)
7       requires //pre-conditions of this program
8           forall|i:int, j:int| 0 <= i <= j < v.len() ==> v[i] <= v[j],
9           exists|i:int| 0 <= i < v.len() && k == v[i],
10      ensures //post-conditions of this program
11          0 <= r,
12          r < v.len(),
13          k == v[r as int],
14  {
15      let mut i1: usize = 0;
16      let mut i2: usize = v.len() - 1;
17      while i1 != i2
18          invariant //loop invariants (used for proof)
19              i2 < v.len(),
20              // exists|i: int| i1 <= i <= i2 && k == v[i],
21              forall|i:  int, j:  int| 0 <= i <= j < v.len() ==> v[i] <= v[j],
22      {
23          let ix = i1 + (i2 - i1) / 2;
24          if v[ix] < k {
25              i1 = ix + 1;
26          } else {
27              i2 = ix;
28          }
29      }
30      i1
31  }
32  }
33
34  Error messages from Verus:
35  error: postcondition not satisfied
36      |
37  13 |            k == v[r as int],
38      |            ---------------- failed this postcondition
39  14 | / {
40  15 | |      let mut i1: usize = 0;
41  16 | |      let mut i2: usize = v.len() - 1;
42  17 | |      while i1 != i2
43  ... |
44  30 | |      i1
45  31 | | }
46      | |_^ at the end of the function body
47
48  error: possible arithmetic underflow/overflow
49      |
50  23 |          let ix = i1 + (i2 - i1) / 2;
51      |                        ^^^^^^^^^^^^^^^^^^^^
52
53  error: possible arithmetic underflow/overflow
54      |
55  23 |          let ix = i1 + (i2 - i1) / 2;
56      |                        ^^^^^^^^^^
57
58  error: aborting due to 3 previous errors
59
60  verification results:: 0 verified, 2 errors
```

### A.1  IMPERFECT PROOF EXAMPLE

An imperfect proof annotation of binary search is shown in Listing 2. Compared with the correct version in Listing 1, this imperfect proof lacks a crucial loop invariant in Line 20. According to the error messages following the code, we can see that a postcondition cannot be proved without this loop invariant. Furthermore, the absence of this invariant causes Verus to fail to prove the absence of arithmetic underflow/overflows. Specifically, this loop invariant states that value k exists in v[i1

Listing 3: The proof code of fibonacci.

```
1   use vstd::prelude::*;
2   fn main() {}
3
4   verus! {
5   spec fn fibo(n: int) -> nat
6       decreases n
7   {
8       if n <= 0 { 0 } else if n == 1 { 1 }
9       else { fibo(n - 2) + fibo(n - 1) }
10  }
11
12  spec fn fibo_fits_i32(n: int) -> bool {
13      fibo(n) < 0x8000_0000
14  }
15
16  proof fn fibo_is_monotonic(i:  int, j:  int)
17      requires
18          i <= j,
19      ensures
20          fibo(i) <= fibo(j),
21      decreases j - i
22  {
23      if i <= 0 {
24      }
25      else if i < j {
26          fibo_is_monotonic(i, j-1);
27          assert(fibo(j) == fibo(j-1)+fibo(j-2));
28      }
29  }
30
31  fn fibonacci(n: usize) -> (ret: Vec<i32>)
32  requires
33      fibo_fits_i32(n as int),
34      n >= 2,
35  ensures
36      forall |i:  int| 2 <= i < n ==> #[trigger] ret@[i] == fibo(i),
37      ret@.len() == n,
38  {
39      let mut fib = Vec::new();
40      fib.push(0);
41      fib.push(1);
42      let mut i = 2;
43
44
45      while i < n
46          invariant
47              forall |k:  int| 0 <= k < i ==> #[trigger] fib@[k] == fibo(k),
48              fibo_fits_i32(n as int),
49              2 <= i,
50              fib@.len() == i,
51              i <= n,
52      {
53          proof{
54              fibo_is_monotonic(i as int, n as int);
55          }
56          let next_fib = fib[i - 1] + fib[i - 2];
57
58          fib.push(next_fib);
59
60          i += 1;
61      }
62
63      fib
64  }
65  }
```

.. i2]. Without this loop invariant, Verus cannot reason about the comparison between v[ix] and k on Line 24, and hence cannot reason about how the values of i1 and i2 would change in this loop. This incapability of reasoning then leads to Verus' failure in reasoning about the bound of i2 − i1 on Line 23 (and hence the arithmetic underflow/overflow error), and the final value of i1 on Line 30 (and hence the postcondition not satisfied error).

## A.2 MORE COMPLICATED PROOF ANNOTATIONS

Listing 3 shows much more complicated proof annotations for a Rust function that computes the Fibonacci sequence. The proof for this code includes not only loop invariants introduced in Listing 1, but also a proof function (Lines 16–29 in the listing), a proof block (Lines 53–55), and `assert` statements (Line 27) that are used to provide hints to the underlying theorem prover. This example comes from the Verus paper (Lattuada et al., 2023), with more detailed explanation available in that paper.

## B PROMPTS

Listing 4: SAFE's prompt for proof generation.

```
1  Instruction: "You are an experienced formal language programmer. You are very familiar with
   Verus, which is a tool for verifying the correctness of code written in Rust. Your mission is
   to write proof code, including loop invariants and assertions to the given Rust code, so that
   Verus can verify the give function behaves exact what is described in the specifications.
   Return the verified code in '''rust''' code block. Here is the given rust code."
2  Input: "'''rust {The Input Rust Program}'''"
```

Listing 5: SAFE's prompt for self-debugging.

```
1  instruction = "You are an experienced formal language programmer. You are very familiar with
   Verus, which is a tool for verifying the correctness of code written in Rust. Your mission is
   to write correct proof code, including loop invariants and assertions to the given Rust code,
   so that Verus can verify the give function behaves exact what is described in the
   specifications, which is 'requires' and 'ensures'. The given verus code cannot be verified,
   there exists errors in the proof code. Please help debug the given code according to the error
    messages. Return the verified code in '''rust''' code block."
2  input = "The given rust is:\n '''rust {The Incorrect Rust
   Program}''', and the error messages are:\n {''' {The Error Messages}'''}.\n"
```

## C IMPLEMENTATION DETAILS

## C.1 SPECIFICATION FILTERING

The example below shows two types of imperfect specification. The specification in the left column is correct but incomplete for the binary search function, as an incorrect implementation that always returns `0` would be accepted by this specification; the specification in the right column below is incorrect, as a correct implementation may fail this specification when the input array has multiple elements matching the search key.

```
1  #A correct, but incomplete specification.        1  #An incorrect specification.
2  0 <= r,                                          2  r > 0 ==> k > v[r - 1],
3  r < v.len(),                                     3  r < v.len() - 1 ==> k < v[r + 1],
```

Listing 7 shows an example of how we leverage test cases and Verus' symbolic reasoning capability to quickly score and filter specifications. For each function with a synthesized specification (i.e., Lines 6–8 in Listing 7, in order to know whether this specification is consistent with a test case $t$, we do the following: 1) We replace the function body with Verus' `assume` statements. Each `assume` tells Verus about the value of one input or output variable (these values come from the test case $t$); 2) When an input/output variable is a container, multiple `assert` statements are inserted to make sure that the underlying theorem prover can correctly reason about the value of every element in the container; 3) Verus is invoked to prove this function. If the specification is consistent with this test case, Verus verification would succeed; otherwise, Verus verification would fail.

Listing 6: GPT-4o's prompt for proof generation.

```
1   system = "You are an experienced formal language programmer. You are very familiar with Verus,
     which is a tool for verifying the correctness of code written in Rust."
2
3   instruction = """
4   Your missions are to
5   1. Add loop invariants to the given Rust code, if there are loops in the code, so that Verus
    can verify the give function behaves exact what is described in the specifications
6   2. Add the proof blocks that could help Verus to prove the following code snippet. You need to
     analyze which locations in the code need to be proved and add the proof blocks to help Verus
    to prove the correctness of the code. You can insert multiple proof blocks in the code as long
     as they are necessary to prove the correctness of the code. You can also include new ghost
    variables that could help you to prove the correctness of the code.
7
8   The proof block looks like this:
9   ```
10  proof {
11      // your proof code here
12      // assert(...)
13      // LEMMA_FUNCTION(...)
14      // ...
15  } // Added by AI
16  ```
17
18  ## Step 1: Add Loop Invariants
19  Please follow these steps in adding loop invariants for every loop:
20  1. You should identify every variable that is read in the loop  (e.g., x[k], y), particularly
    for array elements like x[k], and add an invariant about the initial value for EACH such
    variable and array;
21  2. You should identify every variable that is written (e.g., y = ..., x.set(..,..)) in every
    loop, and add an invariant about the value of that variable. Even if an invariant is already
    specified earlier in the program, please do repeat it in every loop suitable.
22  3. You can leverage the spec functions and proof functions in the invariant.
23
24  ## Step 2: Constant propagation refinement
25
26  If an upper bound or a lower bound about a constant function parameter (e.g., X < ..., X >
    ...) is provided in the function pre-condition (i.e., in the `requires' code block at the
    beginning of the function),
27  please copy that (e.g., X < 10, X > 5) as a loop invariant to every loop in the function.
28  Even if an invariant is already specified earlier in the program, please do repeat it in every
     loop suitable.
29
30  ## Step 3: Array length refinement
31
32  For every loop in the function, please identify every array that is read (e.g., x[k]) or
    written (e.g., x.set(..,..)) in it, and then add a loop invariant that specifies the length of
     the array (i.e., x.len() == ...).
33
34  ## Step 4: Quantifier range refinement
35
36  Please take the following steps to check every loop invariant that involves an array (e.g., x[
    k]) in the given Rust code:
37  If this array x[k] has been modified in this loop through x.set(), leave this invariant as it
    is, do NOT make any changes, and move on to the next invariant.
38  Otherwise, when there is no x.set() in the loop, please make sure that the invariant covers
    every element in the array and hence has the form like `forall |k:int| 0<= k < x.len() ==>
    whatever-property'. When you make this change, please use a comment to explain why you believe
     the related array is never changed in the loop. Do NOT make any other changes to the code or
    the loop invariant!
39
40  ## Step 5: Conditional loop invariant refinement
41
42  Your mission is to refine some loop invariants in the given Rust code only if the loop has
    special handling for the first iteration. This is what you should do: if an existing loop
    invariant P holds for all iterations of the loop except for the first iteration (e.g., some
    variable updates may only (not) occur during the first loop iteration), please leave P as it
    is and add another loop invariant conditioned on the loop index (e.g., index > 0 ==> P),
    following the example below.
43  Do not change P or any other loop invariants in any other way.
44  """
```

As mentioned earlier in the paper, to evaluate the completeness of a specification, we mutate existing test cases to see whether incorrect test cases can be rejected by a specification. In this example, we mutate the output in the ground truth by adding a new value $15$ into $result$ vector. In this case,

Verus will report that the postcondition on Line 7 fails because $15$ does not exist in vector $a$. If a synthesized specification does not contain something like Line 7, this mutated test case will likely point out the incompleteness of that specification.

In our experiments, we take 5 ground truth test cases and 20 mutated wrong test cases on average for scoring. We filter out all the specifications that have a score lower than 0.8 in correctness or 0.6 in incompleteness. To avoid too many specifications being preserved for a single program, we allow at most three specifications to be preserved for a single function.

Listing 7: An example used for scoring and filtering based on test cases.

```
1   use vstd::prelude::*;
2   fn main() {}
3   verus!{
4
5   pub fn SharedElements(a: Vec<i32>, b: Vec<i32>, result: Vec<i32>)
6       ensures
7           forall |k:int| 0 <= k < result.len() ==> (#[trigger] a@.contains(result[k]) && #[
            trigger] b@.contains(result[k])),
8           forall |k1:int,k2:int| 0 <= k1 < k2 < result.len() ==> result[k1] != result[k2],
9   {
10
11      assume(a@ =~= seq![11, 12, 14, 13]);
12      assume(b@ =~= seq![17, 15, 14, 13]);
13      assume(result@ =~= seq![14, 13]);
14
15      assert(a[0] == 11);
16      assert(a[1] == 12);
17      assert(a[2] == 14);
18      assert(a[3] == 13);
19
20      assert(b[0] == 17);
21      assert(b[1] == 15);
22      assert(b[2] == 14);
23      assert(b[3] == 13);
24
25      assert(result[0] == 14);
26      assert(result[1] == 13);
27  }
28  }
```

## C.2 STATISTICS OF SYNTHESIZED DATA

As explained earlier in the paper, we obtain 45,495 Rust single-function programs from the MBPP training set and CodeNet dataset. After translating them into Verus-compatiable Rust programs, we have 21,398 programs left for specification synthesis. The statistics of all synthesized data in different rounds of our self-evolution process are shown in Table 4. Note that, at the very beginning, we put aside a random subset of CodeNet Rust programs to be used for the evaluation of SAFE. Those programs do not participate in the fine-tuning and are not included in the numbers shown in Table 4.

In round 0, we employ GPT-4o to synthesize specifications, retaining only the verified Rust functions along with their specifications. This process is time-consuming, taking an entire month to synthesize 3,673 verified programs. Even with our designed prompts, GPT-4o needs to sample 20–30 times for a single program on average. We then use the synthesized specifications to bootstrap the self-evolving specification synthesis process, which is conducted over two rounds. In each round, we leverage the fine-tuned model to generate specifications for all 21k Verus programs and select the high quality ones for further fine-tuning. At the end, we use the selected 19,017 specifications as inputs for the proof synthesis procedure. The numbers of verified programs and debugging fine-tuning triplets are also shown in Table 4. As we can see, as many as 9,706 verified programs and 10,486 debugging data are produced after the Round-2 model and are used to fine-tune the Round-3 model, being the final proof-synthesis model.

Table 4: Statistics of synthesized data

|  | Verus programs | All specs | Selected specs | Verified programs | Debugging data |
|---|---|---|---|---|---|
| Seed | 21,398 | - | - | - | - |
| Round-0 | - | - | 3,673 | 3,673 | 0 |
| Round-1 | - | 102,549 | 16,530 | 8,368 | 9,910 |
| Round-2 | - | 117,759 | 19,017 | 9,706 | 10,486 |

### C.3 TRAINING HYPERPARAMETERS

Bootstrapped by 3,673 verified programs synthesized by GPT-4o, we employ the DeepSeekCoder model (Guo et al., 2024) as the generator in the self-evolution procedure. We run the self-evolving specification generation and auto-proving process for multiple rounds. For specification generation, we run two rounds. Proof generation is three. At each round of fine-tuning, we combine the verified programs and self-debugging data pairs together as training data, and train DeepSeekCoder 5 epochs, using a batch size of 128 and a learning rate of $1 \times 10^{-5}$.

## D DISCUSSION AND LIMITATION

### D.1 SCALING TO REAL-WORLD LARGE SOFTWARE PROJECTS

In this paper, we have focused on generating proofs for small programs mainly because there are too few existing large Verus projects for us to fine-tune LLMs. Consequently, just like many code-synthesis projects that start from small programs, we also focus on small programs as a starting point for Verus proof synthesis.

Since every function is the unit for Verus verification, we believe that the LLM fine-tuned by SAFE on functions in small programs would continue to be useful for functions in large projects. Of course, if we apply SAFE to synthesize proofs for large Rust projects, we expect a key challenge in how to resolve code dependencies across functions. Specifically, a function may call another executable function or specification function, and the callee function may exist in a different file and/or belong to a different class. How to resolve all the code dependencies and provide LLM with all the needed information may require support that goes beyond machine learning.

### D.2 FINE-GRAINED SELF-DEBUGGING

Figure 2 illustrates that increasing the number of self-debugging rounds hardly improves proof-synthesis accuracy. This situation might be changed if we change the formation of our self-debugging training data. Currently, the training data for self-debugging includes many pairs of incorrect proof $Y_X$ and a corresponding correct proof $Y_\checkmark$. Sometimes, the incorrect proof may contain many mistakes, causing many different verification errors. If future research can break down the difference between $Y_X$ and $Y_\checkmark$, figuring out which edit in $Y_\checkmark$ is used to fix which verification error in $Y_X$, the resulting data can probably train a model that is better at fixing deeply flawed proof through multiple rounds of debugging.

## E ADDITIONAL EXPERIMENTS

### E.1 STATISTICS OF VERUSBENCH

Figure 3 illustrates the distributions of the number of tokens in specifications and proofs in Verus-Bench. For preconditions, in Figure 3a, we show that SV's average number of tokens is substantially larger than that of MBPP and Tutorial. In contrast, SV's average number of tokens is substantially smaller than the rest two as shown in Figure 3b. Such statistics can, to some extent, reflect the difficulty of three components in VerusBench, i.e., SV is relatively easiler than the rest two. When comparing the human-written proof annotations for MBPP and Tutorial programs in VerusBench, Figure 3c shows that the average number of tokens of proof annotations in MBPP is larger than that of Tutorial, although there are some outliers.

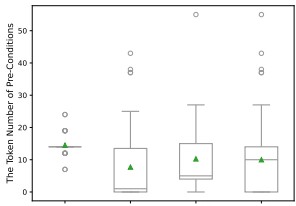 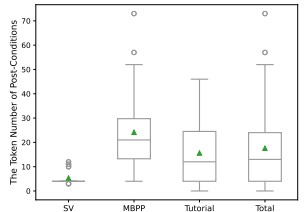 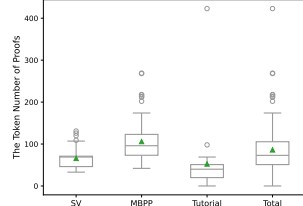

(a) The distribution of the number of tokens in preconditions

(b) The distribution of the number of tokens in postconditions

(c) The distribution of the number of tokens in ground-truth proofs

Figure 3: The statistics of specifications and proofs in VerusBench

## E.2 Distribution of Specification Metrics

Figure 4 shows the distribution of the correctness-score and the completeness-score of all the specifications synthesized during the self-evolving process of SAFE. As we can see, during the self-evolving process, many trivially correct specifications are generated (i.e., $1.0$ correctness score but $< 0.6$ completeness score)—the clusters at the top of the figure and not overlapping with the gray rectangle. These specifications tend to lead to trivial proofs and hence are filtered out. There is also a big cluster of incorrect specifications that have incorrectly rejected all test cases at the bottom-right corner of the figure (i.e., $0$ correctness score but $1.0$ completeness score). There is no way Verus can prove that the Rust function satisfies these specifications, no matter what proof annotations are used, so it is good that SAFE filters all of them out.

## E.3 Results of Statistical Significance Testing

**SAFE Compared to Raw Models.** Table 5 shows the p-values between SAFE under various settings and its base models. We highlight p-values that are larger than 0.05 in italics. From this table, we can observe that the p-values of all fine-tuned models (LLaMa and DeepSeekCoder) are less than 5E-7, indicating the significant improvement of SAFE.

Table 5: The p-values of SAFE's accuracy compared to raw models; non-significant data ($p > 0.05$) points are pointed out in Italics

| Benchmark | Accuracy | GPT-4o | LLaMa3.1-8B-Instruct | | DeepSeekCoder-33B-Instruct | |
|---|---|---|---|---|---|---|
| | | SAFE | SAFE | SAFE+ | SAFE | SAFE+ |
| VerusBench | Accuracy@1 | 2.01E-03 | 4.68E-11 | 3.88E-14 | 2.34E-09 | 6.33E-13 |
| | Accuracy@10 | 4.69E-03 | 5.39E-07 | 5.23E-12 | 5.39E-07 | 1.53E-15 |
| | Accuracy@100 | - | 8.67E-31 | 9.25E-45 | 2.45E-34 | 5.15E-64 |
| CodeNet Test | Accuracy@1 | *0.08* | 2.26E-122 | 1.07E-24 | 1.54E-142 | 1.51E-215 |

**Comparison between Various Rounds of Self-Evolving.** Table 6 shows the p-values of SAFE's accuracy between various rounds of self-evolving. We highlight p-values that are larger than 0.05 in italics. When compared to round 1, round 2 and round 3 show significant improvement in Accuracy, especially after self-debugging. Additionally, we also show that there is insignificant improvement between round 2 and round 3 except the accuracy in Tutorial after self-debugging. Thus, three rounds of self-evolution are sufficient for our approach.

**Comparison between SAFE's Accuracy with Low/High Quality Specifications.** Table 7 shows the p-values of SAFE's accuracy between round 3 and low quality ones. We highlight p-values that are larger than 0.05 in italics. We show that in most scenarios (except Accuracy@1 in round 2 + low quality specifications), low quality specifications might lead to a significant decrease in terms of SAFE's end-to-end effectiveness.

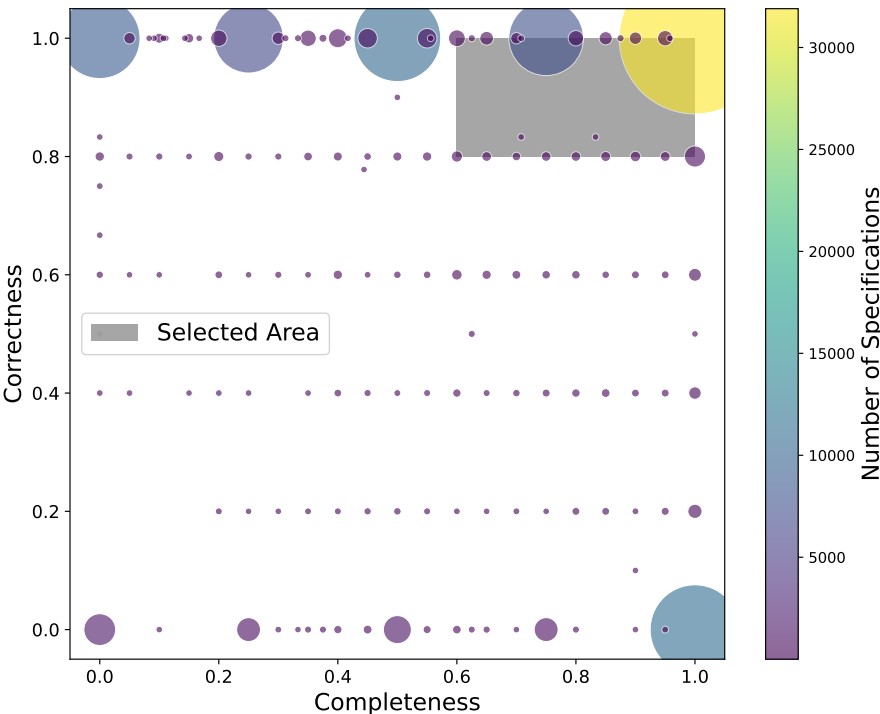

Figure 4: The correctness and completeness distribution of SAFE synthesized specifications. The coordinate of the center of every circle indicates the Completeness-score (x-axis) and the Correctness-score (y-axis) of the corresponding cluster of specifications; the size and color of each circle reflect the number of specifications that are in one cluster. The gray rectangle highlights the range of accepted scores: all the specification clusters that overlap with the gray rectangle are considered good enough and hence kept by SAFE.

Table 6: The p-values of SAFE's accuracy between various rounds of self-evolving; non-significant data ($p > 0.05$) points are pointed out in Italics

| Paired T-Test Rounds | Metric | Proof Generation | | | | Self-Debugging | | | |
|---|---|---|---|---|---|---|---|---|---|
| | | SV | MBPP | Tutorial | Total | SV | MBPP | Tutorial | Total |
| Round 1 & 2 | Accuracy@1 | 1.09E-03 | *1.00* | 4.40E-11 | *0.06* | 6.66E-07 | *0.17* | 2.74E-13 | 2.30E-03 |
| | Accuracy@10 | *0.06* | *1.00* | 9.14E-11 | *0.15* | 7.97E-03 | 1.00E-05 | 5.24E-13 | 3.60E-05 |
| Round 1 & 3 | Accuracy@1 | 2.19E-05 | *0.79* | 3.04E-07 | *0.06* | 1.36E-05 | *0.06* | 3.04E-07 | 4.98E-03 |
| | Accuracy@10 | 7.97E-03 | *0.37* | 5.24E-13 | 4.12E-02 | 1.67E-05 | 1.70E-06 | 1.17E-21 | 9.11E-07 |
| Round 2 & 3 | Accuracy@1 | *0.33* | *0.79* | *0.12* | *1.00* | *0.53* | *0.63* | 2.10E-02 | *0.81* |
| | Accuracy@10 | *0.41* | *0.37* | *0.47* | *0.55* | *0.06* | *0.72* | 2.86E-02 | *0.44* |

### E.4 SELF-EVOLUTION WITH SMALL BACKBONE MODELS

To evaluate the effectiveness of our self-evolution framework on LLMs with a smaller number of parameters, we conduct additional experiments using DeepSeekCoder-1.3B (DSCoder-1.3B) as the backbone model. In this experiment, we follow the same experimental settings described in Section 4.1, bootstrapping DSCoder-1.3B with GPT-4o's predictions.

Table 7: The p-values of SAFE's accuracy with low quality specifications, compared to round 3 with high quality ones; non-significant data ($p > 0.05$) points are pointed out in Italics

| Data Source | Metric | Proof Generation | | | | Self-Debugging | | | |
|---|---|---|---|---|---|---|---|---|---|
| | | SV | MBPP | Tutorial | Total | SV | MBPP | Tutorial | Total |
| Low-Quality | Accuracy@1 | 5.15E-64 | *0.79* | 3.04E-07 | 3.22E-06 | 3.62E-71 | *0.72* | *0.40* | 1.98E-04 |
| | Accuracy@10 | 4.04E-87 | *0.53* | 9.39E-08 | 5.38E-08 | 8.66E-71 | *0.15* | 3.26E-03 | 4.79E-07 |
| Mix-Quality | Accuracy@1 | 1.94E-13 | *1.00* | 8.07E-02 | 2.45E-02 | 1.71E-12 | *0.72* | *0.40* | *0.07* |
| | Accuracy@10 | 2.78E-08 | *0.53* | 2.27E-04 | 3.05E-02 | 5.34E-10 | *0.23* | 2.86E-2 | 2.51E-02 |

Table 8: Each round's accuracy with DeepSeekCoder-1.3B as the backbone model

| Metric | Proof Generation | | Self Debugging ($k + k * k$) | |
|---|---|---|---|---|
| | Accuracy@1 | Accuracy@10 | Accuracy@1 | Accuracy@10 |
| Raw | 1.44 | 6.47 | - | - |
| Round 1 | 12.95 | 24.46 | 12.95 | 24.46 |
| Round 2 | 19.42 | 26.69 | 24.46 | 52.52 |
| Round 3 | 21.58 | 40.29 | 27.34 | 57.55 |

After the same rounds of self-evolving specification generation and proof generation, the results on VerusBench are listed in Table 8. The results indicate that our self-evolution framework effectively improves the accuracy even with a smaller backbone model of 1.3 billion parameters.

# F  CASE STUDIES

## F.1  SAFE'S ADVANTAGES COMPARED TO GPT-4O

To demonstrate the effectiveness of SAFE, we show an example proof generated by GPT-4o without our prompts and SAFE in Listing 8 and Listing 9, respectively.

In Listing 8, GPT-4o introduces four loop invariants, specifying that (1) the index variable `i` does not exceed the length of input array `xlen`; (2) the length of the input array remains during the loop; (3) the values of the first `i` elements are equal to their original values plus 4; (4) the values of the rest elements are equal to their original values. However, GPT-4o neglects to include proofs to address the potential arithmetic underflow/overflow, thus leading to the errors reported in Lines 30-64.

In contrast, SAFE addresses this error and successfully generates a correct proof for this program. Listing 9 shows a correct proof generated by SAFE (DeepSeekCoder). Compared to GPT-4o's proof, it adds another loop invariant in Line 19 pointing out that the original values of the input array are less than or equal to `0x7FFFFFFB`, thus addressing the `arithmetic underflow/overflow` error. SAFE's effectiveness mainly credits to our self-evolution framework.

## F.2  THE EFFECTIVENESS OF OUR SELF-DEBUGGING MECHANISM

In Listing 10 and Listing 11, we demonstrate the effectiveness of our self-debugging mechanism during proof synthesis with another example. This example program checks whether a given array is sorted by checking whether there exists a pair of adjacent elements where `arr[index] > arr[index + 1]`.

Listing 10 shows the initial proof generated by SAFE, including loop invariants: (1) the index does not exceed the length of the given array; (2) all elements before the current index satisfy the `‘‘sorted’’` property. However, during each iteration of this loop, this program checks the current element against the next one and terminates the loop when `index` equals `arr.len() - 2`. As a result, the loop invariants fail to account for the last element, leading to the error described in Lines 29-41.

SAFE successfully fixes this error via self-debugging. The error message clearly indicates that the loop invariant `forall |i:  int, j:  int| 0 <= i < j < index` cannot be verified. Using

Listing 8: An incorrect proof of $myfun2$ generated by GPT-4o.

```
1   use vstd::prelude::*;
2   fn main() {}
3
4   verus!{
5   pub fn myfun2(x: &mut Vec<i32>)
6   requires
7       forall |k:int| 0 <= k < old(x).len() ==> old(x)[k] <= 0x7FFF_FFFB,
8   ensures
9       x@.len() == old(x)@.len(),
10      forall |k:int| 0 <= k < x.len() ==> #[trigger] x@[k] == old(x)@[k] + 4,
11  {
12      let mut i: usize = 0;
13      let xlen: usize = x.len();
14      while i < xlen
15          invariant
16              0 <= i <= xlen,
17              x@.len() == old(x)@.len(),
18              forall |k:int| 0 <= k < i ==> #[trigger] x@[k] == old(x)@[k] + 4,
19              forall |k:int| i <= k < xlen ==> x@[k] == old(x)@[k],
20      {
21          x.set(i, x[i] + 4);
22          i = i + 1;
23      }
24  }
25  }
26  -------------------------------------------------------------------------------
27  note: while loop: not all errors may have been reported; rerun with a higher value for --
    multiple-errors to find other potential errors in this function
28    --> /playground/src/main.rs:14:5
29     |
30  14 | /      while i < xlen
31  15 | |          invariant
32  16 | |              0 <= i <= xlen,
33  17 | |              x@.len() == old(x)@.len(),
34  ... |
35  22 | |          i = i + 1;
36  23 | |      }
37     | |______^
38
39  note: recommendation not met
40    --> /playground/src/main.rs:18:39
41     |
42  18 |          forall |k:int| 0 <= k < i ==> #[trigger] x@[k] == old(x)@[k] + 4,
43     |                                        ^^^^^^^^^^^^^^^^^^
44     |
45    ::: /playground/verus/source/vstd/seq.rs:62:18
46     |
47  62 |              0 <= i < self.len(),
48     |                   - recommendation not met
49
50  ......
51
52  error: precondition not satisfied
53    --> /playground/src/main.rs:21:9
54     |
55  21 |          x.set(i, x[i] + 4);
56     |          ^^^^^^^^^^^^^^^^^^^
57     |
58    ::: /playground/verus/source/vstd/pervasive.rs:378:13
59     |
60  378 |              i < old(self).len(),
61     |              ------------------ failed precondition
62
63  ......
64  error: aborting due to 3 previous errors
```

this feedback from Verus, SAFE modifies the boundary condition in that loop invariant, i.e., the ranges of the variable `i` and `j` in that invariant. The adjusted invariant `forall |i:  int, j: int| 0 <= i < j < index + 1` allows Verus to successfully finish the proof.

Listing 9: The correct proof of $myfun2$ generated by SAFE.

```
1   use vstd::prelude::*;
2   fn main() {}
3   verus!{
4   pub fn myfun2(x: &mut Vec<i32>)
5   requires
6       forall |k:int| 0 <= k < old(x).len() ==> old(x)[k] <= 0x7FFF_FFFB,
7   ensures
8       x@.len() == old(x)@.len(),
9       forall |k:int| 0 <= k < x.len() ==> #[trigger] x@[k] == old(x)@[k] + 4,
10  {
11      let mut i: usize = 0;
12      let xlen: usize = x.len();
13      while (i < xlen)
14          invariant
15              0 <= i <= xlen,
16              x@.len() == old(x)@.len(),
17              forall |k:int| 0 <= k < i ==> #[trigger] x@[k] == old(x)@[k] + 4,
18              forall |k:int| i <= k < xlen ==> x@[k] == old(x)@[k],
19              forall |k:int| 0 <= k < xlen ==> old(x)[k] <= 0x7FFF_FFFB,
20      {
21          x.set(i, x[i] + 4);
22          i = i + 1;
23      }
24  }
25  }
26  ----------------------------------------------------------------------------------
27  verification results:: 2 verified, 0 errors
```

### F.3    AN INEFFECTIVE EXAMPLE

We report that SAFE fails to generate correct proof for 20.86% of the programs in VerusBench, even after sampling 100 proofs and 100 debugged proofs for each incorrect proof (i.e., Accuracy@100). In Listing 12 and Listing 13, we analyze an example where SAFE fails to generate a correct proof. This program removes odd numbers from a given array by scanning the input array `arr` and appending the even numbers to a new array, `even_list`.

Listing 12 shows that SAFE successfully generates the main structure of the proof, verifying that the elements in `even_list` correspond to the even elements in the first `index` elements of `arr`. However, Verus reports errors indicating that the loop invariants are not satisfied both before and after the loop (Lines 34-44 in Listing 12). Even under the Accuracy@100 setting, where 100 * 100 debugged proofs are generated, SAFE encounters similar errors, and fails to generate a correct proof.

The root cause of this error is that proving this program needs to add `assert` statements and leverage the `reveal` feature of Verus (Doc, 2024). We suspect that the `reveal` feature is never synthesized in any finetuning dataset, so the final SAFE model also lacks sufficient domain knowledge to prove this program. This limitation can be substantially mitigated by incorporating a larger and more diverse training set during the initial round of self-evolution.

### F.4    AN EXAMPLE PROOF WITH A LOW-QUALITY SPECIFICATION

In Section 4.3.3, we demonstrate that the quality of specifications contributes substantially to the end-to-end effectiveness of SAFE. Listing 14 presents an example proof with a low-quality specification. This program computes the greatest common divisor of two integer numbers using the Euclidean Algorithm. However, this specification merely requires that the result be greater than zero, ``result > 0'', without ensuring that the `result` should be divisors of both inputs `a` and `b`.

Programs with such low-quality specifications might lead to extremely simple proofs, which are very different from proofs in more realistic tasks. For example, the proof annotations synthesized in Lines 16-17 simply copy the function precondition—`a > 0` and `b >= 0`. For more realistic tasks, proofs can rarely get done this way. Consequently, putting many proofs with this quality (i.e., all resulted from low-quality specifications) into the fine-tuning dataset does not help a model's evolution.

Listing 10: An incorrect proof of $is\_sorted$ generated by SAFE before self-debugging.

```
use vstd::prelude::*;

fn main() {}
verus! {
fn is_sorted(arr: &Vec<i32>) -> (is_sorted: bool)
    requires
        arr.len() > 0,
    ensures
        is_sorted == (forall|i:  int, j:  int| 0 <= i < j < arr.len() ==> (arr[i] <= arr[j])),
{
    let mut index = 0;
    while index < arr.len() - 1
        invariant
            index < arr.len(),
            forall |i:  int, j:  int| 0 <= i < j < index ==> (arr[i] <= arr[j]),
    {
        if arr[index] > arr[index + 1] {
            return false;
        }
        index += 1;
    }
    true
}
} // verus!
----------------------------------------------------------------------------------------------------
Error messages from Verus:
error: postcondition not satisfied
  --> /playground/src/main.rs:10:1
   |
9  |     is_sorted == (forall|i: int, j: int| 0 <= i < j < arr.len() ==> (arr[i] <= arr[j])),
   |     ---------------------------------------------------------- failed this postcondition
10 | / {
11 | |       let mut index = 0;
12 | |       while index < arr.len() - 1
13 | |           invariant
... |
22 | |       true
23 | | | }
   | |_^ at the end of the function body

error: invariant not satisfied at end of loop body
  --> /playground/src/main.rs:15:13
   |
15 |     forall|i: int, j: int| 0 <= i < j < index ==> (arr[i] <= arr[j]),
   |     ^^^^^^^^^^^^^^^^^^^^^^^^^^^^^^^^^^^^^^^^^^^^^^^^^^^^^^^^^^^^^^^^^^

error: aborting due to 2 previous errors

Standard Output
verification results:: 0 verified, 2 errors
```

Listing 11: The correct proof of *is_sorted* fixed by SAFE after self-debugging.

```
1   use vstd::prelude::*;
2
3   fn main() {}
4   verus! {
5   fn is_sorted(arr: &Vec<i32>) -> (is_sorted: bool)
6       requires
7           arr.len() > 0,
8       ensures
9           is_sorted == (forall|i:  int, j:  int| 0 <= i < j < arr.len() ==> (arr[i] <= arr[j])),
10  {
11      let mut index = 0;
12      while index < arr.len() - 1
13          invariant
14              0 <= index <= arr.len() - 1,
15              arr.len() > 0,
16              forall|i:  int, j:  int| 0 <= i < j < index + 1 ==> (arr[i] <= arr[j]),
17      {
18          if arr[index] > arr[index + 1] {
19              return false;
20          }
21          index += 1;
22      }
23      true
24  }
25  } // verus!
26  ------------------------------------------------------------------------------------------
27  verification results:: 2 verified, 0 errors
```

Listing 12: An incorrect proof of *remove_odds* before and after self-debugging.

```
1   use vstd::prelude::*;
2
3   fn main() {}
4   verus! {
5   fn remove_odds(arr: &Vec<u32>) -> (even_list: Vec<u32>)
6       ensures
7           even_list@== arr@.filter(|x:  u32| x % 2 == 0),
8   {
9       let mut even_list: Vec<u32> = Vec::new();
10      let input_len = arr.len();
11
12      let mut index = 0;
13      while index < arr.len()
14          invariant
15              0 <= index <= input_len,
16              // The following is the changed loop invariants before self-debugging.
17              - even_list@== arr@.filter(|x:  u32| x % 2 == 0).take(index).collect(),
18              // The following is the changed loop invariants after self-debugging.
19              + even_list@== arr@.filter(|x:  u32| x % 2 == 0).take(index as int)
20              +                   .filter(|x:  u32| x % 2 == 0),
21      {
22          if (arr[index] % 2 == 0) {
23              even_list.push(arr[index]);
24          }
25          index += 1;
26      }
27      even_list
28  }
29  } // verus!
30  ------------------------------------------------------------------------------------------
31  ......
32  error: invariant not satisfied at end of loop body
33    --> /playground/src/main.rs:19:13
34     |
35  19 |             even_list@ == arr@.take(index as int).filter(|x: u32| x % 2 == 0),
36     |                          ^^^^^^^^^^^^^^^^^^^^^^^^^^^^^^^^^^^^^^^^^^^^^^^^^^^^^
37
38  error: invariant not satisfied before loop
39    --> /playground/src/main.rs:19:13
40     |
41  19 |             even_list@ == arr@.take(index as int).filter(|x: u32| x % 2 == 0),
42     |                          ^^^^^^^^^^^^^^^^^^^^^^^^^^^^^^^^^^^^^^^^^^^^^^^^^^^^^
43  ......
```

The changed loop invariants before and after self-debugging are marked in - red and + green.

Listing 13: The ground truth proof of *remove_odds*.

```
1   use vstd::prelude::*;
2
3   fn main() {}
4   verus! {
5   fn remove_odds(arr: &Vec<u32>) -> (even_list: Vec<u32>)
6       ensures
7           even_list@== arr@.filter(|x:  u32| x % 2 == 0),
8   {
9       let mut even_list: Vec<u32> = Vec::new();
10      let input_len = arr.len();
11
12      assert(arr@.take(0int).filter(|x:  u32| x % 2 == 0) == Seq::<u32>::empty());
13      let mut index = 0;
14      while index < arr.len()
15          invariant
16              0 <= index <= arr.len(),
17              even_list@== arr@.take(index as int).filter(|x:  u32| x % 2 == 0),
18      {
19          if (arr[index] % 2 == 0) {
20              even_list.push(arr[index]);
21          }
22          assert(arr@.take((index + 1) as int).drop_last() == arr@.take(index as int));
23          reveal(Seq::filter);
24          index += 1;
25      }
26      assert(arr@== arr@.take(input_len as int));
27      even_list
28  }
29  } // verus!
30  ------------------------------------------------------------------------------------
31  verification results:: 2 verified, 0 errors
```

Listing 14: An example proof of *gcd* with a low quality specification.

```
1   use vstd::prelude::*;
2   fn main() {}
3
4   verus! {
5   fn gcd(a: u32, b: u32) -> (result: u32)
6       requires
7           a > 0,
8           b >= 0,
9       ensures
10          result > 0,
11  {
12      let mut a = a;
13      let mut b = b;
14      while b != 0
15          invariant
16              a > 0,
17              b >= 0, ,
18      {
19          let temp = b;
20          b = a % b;
21          a = temp;
22      }
23      a
24  }
25  }
26  ------------------------------------------------------------------------------------
27  verification results:: 2 verified, 0 errors
```