# OpenReview forum: "Automated Proof Generation for Rust Code via Self-Evolution"
_ICLR.cc/2025/Conference — ICLR 2025 Poster_

### Official Review · Reviewer_hvY6 · 2024-10-26

**Soundness:** 2
**Presentation:** 2
**Contribution:** 4
**Rating:** 8
**Confidence:** 4

**Summary:**

This paper seeks to finetune a code-generating LLM to generate verification annotations for code.
Specifically, the authors target Verus, which is an SMT-backed automated-theorem-proving-style verifier for Rust code.
The key technical challenge that the authors thus need to overcome is that ths is a very low resource language,
so simple techniques such as finetuning aren't directly applicable; and even API-backed models do so poorly
on this task that naively distilling wouldn't help, either.
Instead, the authors basically bootstrap the finetuning process as follows:
- First, they generate a set of proof *specifications* for some Rust programs using GPT-4o. They then filter out specs which are "low quality", e.g. those which are always true.
- Then, they use these specifications to generate (against using GPT-4, with some expert-crafted task-specific prompts) proof *annotations* for a small subset of these specifications.
- Finally, they bootstrap a finetuning process from these initial annotations, training in each round the open-source model on the correct proofs it generated in the last round.
There's some additional bells and whistles, such as also training on incorrect proofs by framing it as auxiliary repair task, but I believe this summarizes the core idea.

In terms of the experiments, the authors share results both for a small, human-written benchmark and for GPT-transpiled versions of MBPP and SV.
At first glance I was a bit worried about the scale of this data, but given the novelty of the task and the relative lack of Rust datasets in the literature I actually commend the authors on their effort to collect as much data to evaluate on as possible.

**Strengths:**

I think the core ideas in this paper are very interesting, and that there are several good contributions here:
- Filtering the specifications based on a symbolic, deterministic score computed from the tests seems like the right thing to do, and I appreciate the brief ablation study of the impact of this step (section 4.2.4).
- The experiment in 466-474 provide further evidence for previous findings in the code generation literature about "sampling+greedy" self-debugging outperforming "greedy+sampling" (I recommend that the authors consider explicitly comparing these results to e.g. [1, 2]).
- Perhaps most importantly, verifying Rust code is not only potentially impactful but also (as far as I know) a completely novel task; kudos to the authors for going through the effort to collect all the data.


[1]
Teaching Large Language Models to Self-Debug
Xinyun Chen, Maxwell Lin, Nathanael Schärli, Denny Zhou.
International Conference on Learning Representations (ICLR), 2024.

[2]
Is Self-Repair a Silver Bullet for Code Generation?
Theo X. Olausson, Jeevana P. Inala, Chenglong Wang, Jianfeng Gao, Armando Solar-Lezama.
International Conference on Learning Representations (ICLR), 2024.

**Weaknesses:**

There are a few rather major flaws that make me hesitant to recommend the paper for acceptance in its current format.

One is that I find that the paper tries a little bit too hard to sell the novelty of this "SAFE" approach.
Bootstrapping finetuning of LLMs by interleaving it with search has been done before; most people call it "expert iteration" [3] (authors: please correct me if you think there is a *significant* difference between your method and this).
Especially what you call "SAFE_0" feels a bit rich: unless I am mistaken, you are literally just doing synthetic data generation with GPT-4o and filtering the results based on some measure of quality.
Also, on line 359 you say "21,398 [programs] have been successfully transformed [...] by SAFE"; unless I'm mistaken you mean "by GPT-4" here, because at this point you haven't done anything other than asking GPT to first transpile the code to Rust and to then transpile the Rust code into the subset of the language that is supported by Verus.

I would encourage the authors to tone down the language a bit more and focus on the actually novel parts of this paper, which I believe to be: the task target; finetuning on repair tasks to improve generation performance; and the metric used to filter the specification samples.

A more important issue is that I do not think the comparison to the baselines is fair in its current form for the "SAFE+" method.
The authors themselves point out that in this variation (i.e., when you do a round of self-debugging if the initial generation does not succeed), they generate `k * k` repair samples - how can you then compare against pass@1? You have actually drawn `k + k*k` samples from the model, so you should at least compare against a baseline of `pass@(k + k*k)`.
This is an issue that has come up again and again the self-debugging/refinement literature, and I once again encourage the authors to engage with that literature.
You still have good results here - for example, the SAFE+ pass@10 is substantially higher than the SAFE pass@100 on VerusBench - but the way you're currently presenting them overstates their significance.

Finally, the writing could use some more proof reading, especially the abstract and the introduction (but this is a minor complaint).


[3]
@misc{polu2022formalmathematicsstatementcurriculum,
      title={Formal Mathematics Statement Curriculum Learning},
      author={Stanislas Polu and Jesse Michael Han and Kunhao Zheng and Mantas Baksys and Igor Babuschkin and Ilya Sutskever},
      year={2022},
      eprint={2202.01344},
      archivePrefix={arXiv},
      primaryClass={cs.LG},
      url={https://arxiv.org/abs/2202.01344},
}

**Questions:**

- In table 2 you report a "total" column; I noticed the numbers don't add up if you just take the mean of the other columns, so I presume what you're actually doing is taking the mean over all of the samples in the entire dataset? (I think that's what you want to do, I just want to make sure I understood correctly).
- When constructing training data for the repair task, are you doing any filtering to make sure that the "incorrect program" is actually similar to the "correct program", or could they be completely different?
- What is the difference between SAFE and expert iteration, other than your synthetic data generation for the specifications?

---

> ### Author Response · Authors · 2024-11-20
> **Response to Reviewer hvY6**
>
> We sincerely thank Reviewer **hvY6** for the insightful comments.  Our response is the following and our revision in the revised paper is highlighted in blue:
>
>
>
> ### W1. “I find that the paper tries a little bit too hard to sell the novelty of this SAFE approach … expert iteration … I would encourage the authors to tone down the language a bit more”
>
> Thanks for your suggestion about the writing of our paper. We have made the following revision in the revised paper following your suggestion:
>
>
>
> a) Our original submission had a short discussion comparing “expert iteration” and SAFE in Section 2. We have elaborated that discussion in the revised version.
>
>
>
> Basically, indeed, our work applies the self-evolving / expert-iteration approach to a new task:
>
> synthesizing correctness proof for Rust code. This new target task raises some unique challenges and hence require different designs:
>
> - Data scarcity challenge in Verus. We do not have millions of manually-written mathematical theorems for bootstrapping, which is why we had to rely on GPT-4o for bootstrapping. We also have no access to a large quantity of proof problems and hence must synthesize problems (i.e., Rust-code transpilation + specification synthesis) by itself.
>
> - The underlying verification engine. Math-proof synthesis can be decomposed to many small steps or tactics thanks to the interactive theorem prover, LEAN. In contrast, Verus leverages SMT-solver to prove the correctness of a function as a whole. Consequently, step-wise search strategies used in prior work do not apply here. Instead, whole-proof debugging is crucial for SAFE.
>
>
>
> b) We have toned down the language a bit more in Introduction. Particularly, we moved the last paragraph originally in the Introduction, which summarizes the reason-of-success of SAFE and how it can help future research, to the end of Section 2 and rephrased it in the context of prior self-evolving/expert-iteration work.
>
>
>
> c) We replaced some appearance of "SAFE" with more precise phrases (e.g., “GPT-4”) at those places you pointed out.
>
>
>
> ### W2. pass@k metrics for self-debugging is unfair
>
> Thanks for pointing out this issue. In the revised paper, we made several changes to the evaluation section, especially Table 1, to address this fairness concern.
>
> At the end of Section 4.1.2, we explain the number of samples produced under SAFE+
>
> We re-organized Table 1: in the revised version, SAFE+ has no Accuracy@1 result now. Instead, the generate once + debug once setting of SAFE+ is now presented in a newly added row of Accuracy@2, so that it is compared with taking two samples from other non-debugging models in a fair way. Similarly, we moved SAFE+ Accuracy@10 results in our original submission down to the Accuracy@100 row, as that is fairer as suggested by the reviewer.
>
> We have revised the text in Section 4.2.1, so that we present a fairer statement of SAFE+’s capability.
>
> Since Table 2 and Table 3 do not compare SAFE+ with SAFE, instead they are designed to show the differences across rounds and different specification settings, we did not reorganize them in the revised paper. But, we added the annotation of “(K + K*K)” right after the “Self-Debugging” column name to remind readers that SAFE+ produces many more than K samples.
>
>
>
>
>
> ### Q1. Table 2’s “total” column
>
> The “total” column is the mean Accuracy over all of the proof tasks in the entire VerusBench dataset. They are not the mean of the other columns in Table 2, because there are different numbers of proof tasks in SV, MBPP, and Tutorial.
>
>
>
> ### Q2. “When constructing training data for the repair task, are you doing any filtering to make sure the ‘incorrect program’ is actually similar to the ‘correct program’?”
>
> We do not filter based on the similarities between “incorrect program” and “correct program”.   For each proof task (i.e., a Rust program with a specification), we first divide its incorrect proofs into subgroups where each group's proofs have the same number of verification errors and then randomly sample at most 10 errors in each subgroup. Our target is to ensure the diversity of debugging data for training.
>
>
>
> ### Q3. “the difference between SAFE and expert iteration”
>
> Please refer to the response to W1 and Section 2 in the revised paper.

---

> > ### Comment · Reviewer_hvY6 · 2024-11-20
> >
> > I thank the authors for their extensive reply to my comments and for successfully addressing all of my concerns.
> >
> > I have reviewed the changes made to the paper, as well as the replies to the other reviews. After doing so I am confident that this paper would make for a good contribution to the conference, and I have updated my score accordingly.

---

### Official Review · Reviewer_K5mr · 2024-11-01

**Soundness:** 4
**Presentation:** 4
**Contribution:** 4
**Rating:** 8
**Confidence:** 4

**Summary:**

The presented paper proposes a method to bootstrap training data for generating proofs of Rust code using LLMs. The pipeline starts from a small set of curated programs and gradually evolves using the verifier as signal for dataset curation. Finally they evaluate the resulting fine-tuned LLM and show state of the art results on a difficult dataset of low-resource correctness proofs.

**Strengths:**

- The paper is well-written and nicely structured. The figures and tables are well formatted and legible.
- The story is engaging and the tackled topic highly relevant.
- The results are clearly presented and provide interesting insights.

**Weaknesses:**

- In Table 1 the Accuracy of GPT-4o on VerusBench @100 is unfortunately missing (likely due to high inference cost?). Similarly the result of DeepSeekCoder RAW @100 is missing. If the authors could provide these values, the tables would provide a much more complete picture.
- In Table 2, Round 3 appears to severely degrade performance of the resulting model on the Tutorial dataset. Does this constitute some first signs of overfitting or collapse or could the authors provide some more insight on what is happening here? It might be interesting to provide some basis on deciding where to stop the iterative process.
- There is no discussion of Limitations. While the provided method is clearly powerful some discussion on potential limitations would be highly appreciated.

**Questions:**

Please provide a short statement or clarification to the points raised above.

---

> ### Author Response · Authors · 2024-11-20
> **Response to Reviewer K5mr**
>
> We sincerely thank the insightful comments from Reviewer **K5mr**. The following is our response, and we have highlighted our revisions in blue in the revised paper:
>
>
>
> ### W1. “In Table 1 the Accuracy of GPT-4o on VerusBench@100 is unfortunately missing. Similarly, the result of DeepSeekCoder RAW @100 is missing”
>
> Thanks for pointing out this issue. We have added the evaluation results of the Accuracy@100 on two raw models and GPT-4o in Table 1. Naturally, these models are all able to prove more tasks with 100 tries, comparing with only 10 or fewer tries. However, even with 100 tries, they still perform much worse than SAFE models with 10 or fewer tries even without SAFE’s self-debugging feature.
>
>
>
> ### W2. “In Table 2, Round 3 appears to severely degrade performance of the resulting model on the Tutorial dataset”
>
> The degradation only occurs for Accuracy@1 for the Tutorial dataset. The main reason is that proofs generated by LLMs might be vulnerable to minor issues --- e.g., using an integer type with bit-width (i32) instead of an integer type without bit-width (int) may cause the whole proof to break down. The occurrence of these issues is rather random and typically goes away when more samples are synthesized. For example, if we look at Accuracy@10, Round 3 model’s accuracy is consistently better than that of Round 2 for every part of VerusBench, as shown in Table 2.
>
>
>
> Our iterative process should stop when the accuracy improvement becomes marginal. As shown in Table 6, the improvement between Round 2 and Round 3 is not significant in most settings, so we stop our self-evolution at Round 3.
>
>
>
> ### W3. “There is no discussion of Limitations”
>
> We have added a “Discussion and Limitation” section in Section D (in Appendix). Specifically, we explain two limitations of our SAFE approach and potential future work, scaling to real-world software projects and designing a fine-grained self-debugging strategy.

---

> ### Comment · Reviewer_K5mr · 2024-11-20
> **Thank you for the Response**
>
> Thank you for providing the requested additional details. I will further read through points raised in other reviews but don't expect to adjust my score.

---

### Official Review · Reviewer_aKrU · 2024-11-03

**Soundness:** 3
**Presentation:** 3
**Contribution:** 3
**Rating:** 6
**Confidence:** 4

**Summary:**

This paper introduces SAFE, an innovative framework designed to address the challenges of automated proof generation for Rust code, SAFE overcomes the significant data scarcity issue ( i.e., there is far less proof data than code data for training language models) by using a self-evolving approach to synthesize specification/proof data and fine-tune models iteratively. SAFE operates through a feedback loop where synthesized proofs are continuously verified by a symbolic verifier, distinguishing correct from incorrect proofs. Incorrect proofs are used to train the model's self-debugging ability, while the correct proofs are used to improve the model for the next round. The design of the approach is smart and uses the insight that (1) using a quantitative metric to select high-quality specifications for fine-tuning; (2) we only need reasonably well, instead of perfect specifications to fine-tune in the next step; and (3) Verus can quickly tell correct proof from incorrect ones, which enables the collecting and filtering of large amount of data.

SAFE achieves a substantial improvement, attaining a 70.50% accuracy on a benchmark set crafted by human experts, a notable advancement over GPT-4o's performance of 24.46%. SAFE also obtains self-debugging ability using the incorrect proofs collected during the data collection step. Experiments show that each round of self-evolving improves the accuracy of SAFE, and proves the importance of using high-quality training data.

**Strengths:**

1. This paper proposes a novel approach that use self-evolving to iteratively improve LLM's ability of generating Rust specification and proofs. The fact that this approach does not rely on larger LLM such as GPT-4o in the following iterations (except the first round) makes it more generalizable and scalable.
2. The proposed approach shows great effectiveness, with three round of self-evolving, the fine-tuned LLM shows about 40% higher accuracy@1 compared to the prompting approach.
3. Comprehensive analysis and experiments, showing that each round of 1, 2 and 3 brings some improvement to the fine-tuned LLM (although the round 2 model is better than the round 3 model under some settings), and showing that high-quality specifications important to improve the model's accuracy during self-evolving.

**Weaknesses:**

1. The self-debugging ability is shown to be only effective for the first time, what could be potential approach for improving the self-debugging ability in the following rounds?
2. I am wondering if this self-evolving approach can improve smaller LLMs ability. For instance, if the backbone is DeepSeekCoder-1.3B, how effective is the self-evolving approach?

**Questions:**

1. A clarifying question about the self-evolving data: The data collected through GPT-4o (round 0) is used to fine-tuned the first specification/proof generation model. What's the data input used to let the generation model generate data for then next round?
* Are these data the same programs as those used in the generating round 0 data? If this is the case, would the training data in each round kind of repetitive and lack of diversity?
* Or does author use some strategies to leave some unique programs for each round, so that the fine-tuning data for each round contains different programs?
2. Self-debugging is quite effective and improves the accuracy, how does the model obtain the ability of self-debugging? does the fine-tuning procedure contains self-debugging training data?
3. Why are the baseline models prompted with 4 examples instead of more examples?

---

> ### Author Response · Authors · 2024-11-20
> **Response to Reviewer aKrU**
>
> We sincerely thank Reviewer **aKrU** for the insightful comments.  We provide our response below and highlight our revisions in blue in the revised paper:
>
>
>
> ###  W1.“What could be potential approach for improving the self-debugging ability in the following rounds”
>
> This is a very good question, and definitely worth future research to look into.
>
> We feel the potential approach likely needs to change the formation of our current self-debugging training data. Currently, the training data for self-debugging includes many pairs of incorrect proof Yx and a corresponding correct proof Y. Sometimes, the incorrect proof Yx
>
> may contain many mistakes, causing many different verification errors. If future research can break-down the difference between Yx and Y, figuring out which edit in Y is used to fix which verification error in Yx, the resulting data can probably train a model that is better at fixing deeply flawed proof through multiple rounds of debugging.
>
> We have added this discussion into the revised paper (Section D.2 in Appendix).
>
>
>
> ###  W2.“ SAFE’s ability on smaller LLMs”
>
> We have conducted a new experiment with DeepSeekCoder-1.3B as backbone (Section E.4 and Table 8 in Appendix).
>
> We use the same experimental setting as in the original submission on bigger LLMs, bootstrapping DSCoder-1.3B with GPT-4o's predictions.
>
> After the same rounds of self-evolving specification generation and proof generation. The results on the VerusBench are as below.
>
> |         | Proof Generation |             | Self Debugging (k+k*k) |              |
> |---------|------------------|-------------|------------------------|--------------|
> | Metric  | Accuracy@1       | Accuracy@10 | Accuracy@1             | Accuracy@10  |
> | raw     | 1.44             | 6.47        | -                      | -            |
> | Round 1 | 12.95            | 24.46       | 12.95                  | 24.46        |
> | Round 2 | 19.42            | 26.69       | 24.46                  | 52.52        |
> | Round 3 | 21.58            | 40.29       | 27.34                  | 57.55        |
>
> The results demonstrate that even when the model size is small, our self-evolution approach can still improve its capability of proof generation.
>
>
>
> ###  Q1.“ What's the data input used to let the generation model generate data for the next round? … Are these data the same programs as those used in the generating round 0 data? Would the training data in each round kind of repetitive and lack of diversity”
>
>
> When SAFE trains the proof-generation model, all the proof tasks (each proof task is a Rust function associated with a specification) that have *not yet* been proved by earlier rounds’ models are input used to let the generation model produce data. If any previously unproved task is now proved by the latest model, the proof is then added to the training set for the next round.
>
>
> In each round, we fine-tune our model based on the raw model, e.g., DeepSeekCoder-33b, using all the correct proofs generated so far in all previous rounds. Following the discussion above, if Round k manages to prove many proof tasks that were not proved in earlier rounds, the training data for Round K+1 will be much richer than the training data used for Round-k and all the earlier rounds. On the other hand, if Round-k only manages to prove few tasks not proved before, the training data for Round k+1 will indeed be kind of repetitive comparing with the training data used for Round k. In that case, the self-evolving process should stop.
>
> When SAFE trains the spec-generation model, the situation is a little bit different. At each round, all the 21K Rust programs are data input used to let the spec-generation model generate specifications. This part of the details is discussed in Section C.2 in the Appendix.
>
> ### Q2. “Self-debugging is quite effective and improves the accuracy, how does the model obtain the ability of self-debugging? does the fine-tuning procedure contains self-debugging training data?”
>
> In our proof-generation step, we fine-tune our model on two tasks simultaneously, proof generation and self-debugging; the self-debugging training data for each round comes from the correct and incorrect proofs synthesized by our models in earlier rounds. We have revised Section 3.3 (self-evolving proof-synthesis) to make this clear, together with formal definitions of our two fine-tuning tasks.
>
> ### Q3. “Why are the baseline models prompted with 4 examples instead of more examples?”
>
> These 4 examples have included the main language features of Verus proof annotations, and hence are sufficient for GPT-4o to conduct in-context learning. When we designed the GPT-4o prompt, we found that adding more examples does not clearly improve GPT-4o’s output quality. Furthermore, more examples, which means longer context, would lead to longer GPT-4o inference cost and time, which we cannot afford --- our current bootstrapping round already requires one month of non-stop GPT-4o invocation.

---

> > ### Comment · Reviewer_aKrU · 2024-11-26
> >
> > Thank you for the detailed response and the additional experiments using DeepSeek-Coder-1.3B. I plan to keep my score.

---

### Official Review · Reviewer_73u5 · 2024-11-03

**Soundness:** 3
**Presentation:** 2
**Contribution:** 3
**Rating:** 6
**Confidence:** 4

**Summary:**

This paper proposes SAFE, a data generation and fine-tuning procedure for improving LLMs in generating proofs for the correctness of Rust code. SAFE consists of three stages: (i) verus-compatible code generation, (ii) self-evolving specification synthesis, and (iii) self-evolving proof synthesis. During stage (ii), SAFE leverages a symbolic and quantitative measure based on the correctness and completeness of the specification. For stage (iii), SAFE fine-tunes both proof generation and repair models. The experiments demonstrate the advantages of SAFE: it significantly improves the performance, compared to both the base model and GPT-4o.

**Strengths:**

This paper studies automating proof generation in formal program verification with LLMs, an important direction with great potential for practical applications. The focus is on Rust, a relatively new language that is gaining widespread adoption. Although synthetic data generation for fine-tuning LLMs is not a completely novel idea, the paper introduces a few interesting techniques for the domain of proof generation for Rust. I particularly like the metric for filtering high-quality specifications. The evaluation is thorough, demonstrating the benefits of SAFE over baselines and the effectiveness of its individual components.

**Weaknesses:**

The paper only focuses on small programs in the style of MBPP and CodeNet. Although I understand this is partly due to the limitation of the Verus tool, I do believe that the paper should present some case studies or discussion on how to scale the approach to real-world software projects.

Apart from proof generation, a major part of formal verification is writing the specifications. The paper covers mechanisms to fine-tune a “good” specification generation. It would strengthen the paper if more evaluation can be done on the specification generation task and how it can be combined with proof generation to automate end-to-end verification.

The paper lacks a study on the choice of the correctness and completeness thresholds for the specification metric.

The paper writing can be improved. Below are some issues I found or some recommendations:
- The text in Section 3 is sometimes ad-hoc and contains low-level details (e.g., choice of parameters). I would be helpful to revise the text to be more formal and move the details to later sections.
- Line 289: The paper says “much previous work relies on running many test cases” without providing any references.
- Line 519: Table 2 should be Table 3
- Table 3: The split of model names to multiple lines is confusing. I thought one line of text corresponds to one single baseline. The $\Delta$ rows look redundant as well.

**Questions:**

Please address the points raised in the “Weakness” section.

---

> ### Author Response · Authors · 2024-11-20
> **Response to Reviewer 73u5**
>
> We sincerely thank Reviewer **73u5** for the insightful comments.  We provide our response below and highlight related revisions in blue in the revised paper:
>
>
>
> ### W1. “discussion on how to scale the approach to real-world software projects”
>
> We have added a discussion in Section D.1 (in Appendix) on this topic.
>
>
>
> Since every function is the unit for Verus verification, we believe the LLM fine-tuned by SAFE on functions in small programs would continue to be useful for functions in large projects. Of course, if we apply SAFE to synthesize proof for large Rust projects, we expect a key challenge in how to resolve code dependencies across functions: a function may call other executable functions or specification functions, and the callee functions may exist in a different file and/or belong to a different class. How to resolve all the code dependency and provide LLM with all the needed information may require support that goes beyond machine learning.
>
>
>
> ### W2.“It would strengthen the paper if more evaluation can be done on the specification generation task and how it can be combined with proof generation to automate end-to-end verification”
>
> We have added further evaluation results and related discussion in Section E.2 and Figure 4 (in Appendix of the revised paper) that show the distribution of the correctness-score and the completeness-score of all the specifications synthesized during the self-evolving process of SAFE.
>
>
>
> In our original submission, we designed two baselines to show how the quality of specification generation would affect the end-to-end verification and hence the effectiveness of our proof generation training: this result was shown in Table 3 (and P-values in Table 7). As the reviewer pointed out, our original presentation in Table 3 was unclear; so, we have cleaned up Table 3 in the revised paper. In general, Table 3 shows that the quality-decrease in specification substantially decreases the effectiveness of end-to-end verification for the Rust programs in our training dataset and hence the accuracy of the final proof-generation model.
>
>
>
> ### W3. “The paper lacks a study on the choice of the correctness and completeness thresholds for the specification metric”
>
> As explained in Section 3.2, SAFE needs specifications that have reasonably high scores, but not perfect scores (i.e., 1.0). Beyond the reasons that are already presented in Section 3.2, an extra reason for the relatively low Completeness threshold (0.6) is that a mutated test case might still be correct and hence should not be rejected. For example, Listing 7 illustrates a test case for “sharedElements” while it does not require the order of output list to be the same as its inputs. If we change the target output from [13, 14] to [14, 13], it is still correct and hence will lower the Completeness score of some good specifications.
>
>
>
> We apologize that we did not have time to re-run the whole training process using different specification-filtering thresholds. We hope that the newly added Figure 4 in the revised paper (it shows the score-distribution of synthesized specifications) and the newly cleaned-up Table 3 (how proof-synthesis accuracy drops when different sets of specification are used) will help readers see how our specification filtering has helped SAFE.
>
>
>
> ### W4. Writing issues
>
>
>
> **4.1 “The text in Section 3 is sometimes ad-hoc and contains low-level details”**
>
> In the revised paper, we have deleted some ad-hoc discussion in Section 3, moved some ad-hoc discussion into Appendix (e.g., the now early part of Section C.1 Specification Filtering), and added formal definitions about our task target and specification metrics (Formula (1), (2), and (3) in Section 3).
>
>
>
> **4.2 “The paper says `much previous work relies on running many test cases’ without providing any references”**
>
> We have added reference to this sentence. It is Lines 271-272 now.
>
>
>
> **4.3 “Table 2 should be Table 3”**
>
> We have changed the incorrect reference. It is Line 521 now.
>
>
>
> **4.4 “Table 3: The split of model names to multiple lines is confusing … The delta rows look redundant as well”**
>
> We have changed the model names in Table 3 and removed $\Delta$ rows.

---

> > ### Comment · Reviewer_73u5 · 2024-11-22
> >
> > Thank the authors for preparing a detailed rebuttal! I have read them but plan to keep my score.

---

### Author Response · Authors · 2024-11-20
**General Response on Paper Revisions**

We sincerely thank all the reviewers for their insightful and valuable comments. We have revised our paper based on the suggestions. They are marked in blue in the newly submitted paper.

- Suggested by Reviewer **hvY6**, in Abstract and Introduction, we remove the last paragraph of Section 1 and tone down the language a bit more to avoid overclaim the novelty of SAFE.

- Suggested by Reviewer **hvY6**, in Section 2, we elaborate the difference from expert iteration.

- Suggested by Reviewer **73u5** and **hvY6**, in Section 3.2 and 3.3, we move the low-level details to Appendix and revise the text to be more formal.

- Suggested by Reviewer **K5mr** and **hvY6**, we re-organize Table 1 and revise the corresponding description in Section 4 for, 1) filling in the missing results of Accuracy@100 for baseline models. 2) Add Accuracy@2 results to make a fair comparison between debugging approach SAFE+ and other methods without debugging.

- Suggested by Reviewer **73u5**, **aKrU** and **K5mr**, we add a new Section in Appendix D to discuss the limitations of our approach. Specifically, we discuss how to scale SAFE to real-world software projects (weakness 1 of Reviewer 73u5) and how to improve the self-debugging training beyond one round (weakness 1 of Reviewer aKrU).

- Suggested by Reviewer **73u5**, we add more evaluation on specifications in Appendix E.2.

- Suggested by Reviewer **aKrU**, we add new experimental results on the effectiveness of SAFE with smaller models in Appendix E.4.

- We add some missing citations and revise some expressions as suggested by the reviewers.

---

### Meta-Review · Area_Chair_MrJn · 2024-12-19

**Metareview:**

This paper considers learning how to prove the correctness of Rust programs, an important programming language for which there is very little training data, by bootstrapping both specifications and proofs. This problem is more broadly emblematic of the need to produce verified code for low resource languages, potentially even low resource areas of math (though not explored in the paper). The primary strength is that the method is creative, potentially high impact, and has good empirical results. The primary weakness is that it only considers proofs for MBPP-style problems, and required nontrivial manual effort to bootstrap the system (while also relying on automated general-purpose methods). I recommend acceptance because this weakness is understandable given that this would only be the first step in this research program, and because it is both conceptually interesting and practically relevant to machine learning and formal methods.

**Additional Comments On Reviewer Discussion:**

Reviewers engaged in the discussion and suggested drawing more connections to existing work. The text has been revised to take such connections into account.

---

### Decision · Program_Chairs · 2025-01-22

Accept (Poster)